# Lytic transglycosylases mitigate periplasmic crowding by degrading soluble cell wall turnover products

Anna Isabell Weaver[1,2], Laura Alvarez[3], Kelly M Rosch[1], Asraa Ahmed[1,4], Garrett Sean Wang[1], Michael S van Nieuwenhze[5,6], Felipe Cava[3]*, Tobias Dörr[1,2,4]*

[1]Weill Institute for Cell and Molecular Biology, Cornell University, Ithaca, United States; [2]Department of Microbiology, Cornell University, Ithaca, United States; [3]The Laboratory for Molecular Infection Medicine Sweden (MIMS), Department of Molecular Biology, Umeå University, Umeå, Sweden; [4]Cornell Institute of Host-Microbe Interactions and Disease, Cornell University, Ithaca, United States; [5]Department of Molecular and Cellular Biochemistry, Indiana University, Bloomington, Sweden; [6]Department of Chemistry, Indiana University, Bloomington, United States

*For correspondence:
felipe.cava@umu.se (FC);
tdoerr@cornell.edu (TD)

**Competing interest:** The authors declare that no competing interests exist.

**Abstract** The peptidoglycan cell wall is a predominant structure of bacteria, determining cell shape and supporting survival in diverse conditions. Peptidoglycan is dynamic and requires regulated synthesis of new material, remodeling, and turnover – or autolysis – of old material. Despite exploitation of peptidoglycan synthesis as an antibiotic target, we lack a fundamental understanding of how peptidoglycan synthesis and autolysis intersect to maintain the cell wall. Here, we uncover a critical physiological role for a widely misunderstood class of autolytic enzymes, lytic transglycosylases (LTGs). We demonstrate that LTG activity is essential to survival by contributing to periplasmic processes upstream and independent of peptidoglycan recycling. Defects accumulate in *Vibrio cholerae* LTG mutants due to generally inadequate LTG activity, rather than absence of specific enzymes, and essential LTG activities are likely independent of protein-protein interactions, as heterologous expression of a non-native LTG rescues growth of a conditional LTG-null mutant. Lastly, we demonstrate that soluble, uncrosslinked, endopeptidase-dependent peptidoglycan chains, also detected in the wild-type, are enriched in LTG mutants, and that LTG mutants are hypersusceptible to the production of diverse periplasmic polymers. Collectively, our results suggest that LTGs prevent toxic crowding of the periplasm with synthesis-derived peptidoglycan polymers and, contrary to prevailing models, that this autolytic function can be temporally separate from peptidoglycan synthesis.

## Editor's evaluation

This study addresses a major missing element in the understanding of how bacteria grow their cell wall and the role of lytic transglycosylases in this process. It had been previously assumed these enzymes cut glycan strands to make room for the insertion of new glycans. However, results presented in this manuscript demonstrate these enzymes have a very different, yet essential role in degrading uncrosslinked glycan strands in the periplasm. The authors further demonstrate that in the absence of lytic transglycosylases cells undergo periplasmic stress due a toxic accumulation of these 'free strands' in the periplasm. The work will be of interest to those in the bacterial growth and division field.

## Introduction

The bacterial cell wall is a nearly universal feature of the bacterial cell envelope. Made primarily of the strong and elastic polymer peptidoglycan (PG), the cell wall preserves bacterial shape while protecting the cell from its high internal turgor pressure and external environmental challenges (*Osawa and Erickson, 2018*; *Cabeen and Jacobs-Wagner, 2005*; *Egan et al., 2020*; *Zhao et al., 2017*; *Sochacki et al., 2011*). PG synthesis begins in the cytoplasm with the generation of lipid II, consisting of a disaccharide (N-acetylmuramic acid [MurNAc]-N-acetylglucosamine [GlcNAc]) that is modified with a pentapeptide side stem and attached to the lipid carrier undecaprenol. Lipid II is flipped across the cytoplasmic membrane where the cell wall is assembled in two reactions: first, lipid II is polymerized into longer glycan strands by glycosyltransferases (GTases) followed by crosslinking of the elongating PG strands via their peptide side stems by transpeptidases (TPases) (*Egan et al., 2020*; *Zhao et al., 2017*; *Cho et al., 2016*; *Leclercq et al., 2017*; *Taguchi et al., 2019*). Ultimately, the combined result of GTase and TPase activities is a covalently closed mesh-like macromolecular network called the PG sacculus.

The strength of the PG sacculus is a double-edged sword. On the one hand, the covalent network provides a mechanical structure strong enough to withstand the high cellular turgor pressure and stresses of a changing environment. On the other hand, it acts as a macromolecular cage that might inhibit cellular expansion and division as well as the insertion of crucial trans-envelope machinery (i.e., flagella and pili) (*Nambu et al., 1999*; *Santin and Cascales, 2017*; *Höltje, 1998*; *Atassi, 2017*; *Viollier and Shapiro, 2003*; *Uehara and Park, 2008*). Bacteria therefore need to couple new PG synthesis with degradation of bonds within the PG network to simultaneously maintain the integrity of the sacculus while also making space for the insertion of new PG (*Höltje, 1998*; *Nguyen et al., 2015*; *Lee and Huang, 2013*).

PG degradation is accomplished by several divergent enzyme classes that are often subsumed under the term 'autolysins,' that is, enzymes that cleave various bonds within PG. Endopeptidases (EPs), for example, cleave the peptide crosslinks and are particularly integral to the 'space-making' autolytic function that permits sacculus expansion during cell elongation; without EP activity, PG synthesis results in a thicker cell wall or integrity failure and lysis (*Murphy et al., 2021*; *Singh et al., 2012*). Another major class of autolysins, the lytic transglycosylases (LTGs), cleave the glycosidic linkages between disaccharide subunits within PG strands. Their biochemistry has been exquisitely well-studied and the diversity of their structures and mechanisms of action well-characterized (*Byun et al., 2018*; *Dik et al., 2017*; *Lee et al., 2018*; *Vijayaraghavan et al., 2018*). Unlike the other autolysins, the primary cleavage mechanism of LTGs is non-hydrolytic. Rather, LTGs perform an intramolecular cyclization of MurNAc residues to generate a unique and readily identifiable signature of their activity, anhydro-MurNAc (anhMurNAc) (*Dik et al., 2017*; *Höltje et al., 1975*; *Williams et al., 2018*). Early characterization of PG in the bacterial sacculus suggested that every PG strand terminates in an anhMurNAc cap, implicating LTGs as potential 'terminases' of GTase glycan elongation (*Höltje et al., 1975*; *Kraft et al., 1998*). This was recently confirmed empirically with the novel discovery and characterization of MltG and its functional analogs (*Bohrhunter et al., 2021*; *Yunck et al., 2016*; *Sassine et al., 2021*; *Tsui et al., 2016*). *Escherichia coli* MltG associates with active PG synthetic complexes to release new strands from the cytoplasmic membrane (to which they are initially tethered via undecaprenyl pyrophosphate), presumably as they emerge from GTase activity; consequently, MltG is a strong determinant of PG strand length (*Bohrhunter et al., 2021*; *Yunck et al., 2016*; *Sassine et al., 2021*; *Tsui et al., 2016*). Whether association with active GTases is a conserved characteristic of MltG functional analogues (including some hydrolytic glycosidases; *Taguchi et al., 2021*) has not been established, so these enzymes may be more broadly referred to as 'PG release factors' since their cleavage of membrane-bound PG may not directly influence PG elongation. Other physiological roles assigned to LTGs include local PG editing for insertion of PG-spanning protein complexes (*Santin and Cascales, 2017*; *Atassi, 2017*; *Koraimann, 2003*) and PG recycling. PG recycling, the reincorporation of PG breakdown products into the biosynthesis cycle, starts with the uptake of PG turnover products by the importer AmpG (*Jacobs et al., 1994*). AmpG specifically imports LTG breakdown products (anhMurNAc-containing fragments), and this process can theoretically be supported by any active LTG that produces monomeric anhydromuropeptides. Despite being well-conserved across many bacterial phyla (and some chloroplasts), PG recycling is not an essential process under standard growth conditions (*Dik et al., 2018*). Another underappreciated function of LTGs has emerged through the work

of our group and others wherein certain individual or combinatorial LTG mutations, including septal LTG RlpA, result in a daughter cell separation defect (*Heidrich et al., 2002*; *Jorgenson et al., 2014*; *Weaver et al., 2019*; *Priyadarshini et al., 2006*). Intriguingly, the roles of LTGs in all these functions – PG release from the membrane, PG recycling, insertion of PG-spanning complexes, and daughter cell separation – do not appear to be essential to bacterial growth in the contexts studied to date.

The apparent nonessentiality of LTG activity would seem contradictory to this enzyme class's broad conservation, as well as the genetic and functional redundancy within many individual species – a trait indicative of an important, conserved function (*Yunck et al., 2016*). Yet answering the most general physiological questions about LTGs has been severely encumbered by LTG redundancy, as rarely does a single LTG mutation yield a significant, readily investigable phenotype. We recently showed that collectively, LTG activity indeed seems to be essential for growth and division (*Weaver et al., 2019*). The actual physiological function for the majority of LTGs, however, and the reason for their collective essentiality, has remained elusive. Here, we sought to comprehensively illuminate physiological roles for LTGs by extensively characterizing mutants of the Gram-negative pathogen *Vibrio cholerae* that are defective for most or all of the species' eight currently annotated LTGs. We find that the vast majority of LTGs are dispensable for growth in laboratory media. Minimal LTG strains are defective for growth in low-salt media and hypersensitive to accumulation of periplasmic sugar polymers. Through analysis of PG turnover products, we show that soluble PG strands accumulate in the wild-type and that this is exacerbated in ΔLTG mutants and alleviated by inactivating a major PG EP. Taken together, our data suggest that LTG activity downstream of PG synthesis mitigates toxic periplasmic accumulation of uncrosslinked, polymeric PG turnover products released by EPs.

## Results

### A single LTG is necessary and sufficient for *V. cholerae* growth

We recently reported that a Δ6 LTG mutant (*rlpA⁺ mltG⁺ ΔmltA ΔmltB ΔmltC ΔmltD ΔmltF Δslt70*) was viable under standard laboratory conditions and exhibited only slight morphological defects, including an increase in cell length (*Weaver et al., 2019*). Depletion of RlpA from this background resulted in a lethal chaining defect, suggesting that collectively, some degree of LTG activity is essential for *V. cholerae* growth. Since we were previously unable to delete MltG or observe its depletion from a Δ6 LTG background, we asked whether MltG exhibited a synthetic-lethal relationship with the other LTGs using a quantitative insertion/disruption assay. Briefly, we conjugated a suicide vector targeting *mltG* (or positive or negative control loci) into WT and Δ6 LTG and quantified viable recombinants. Surprisingly, the WT and Δ6 LTG strains both tolerated inactivation of *mltG* by the suicide vector so long as essential DNA synthesis genes downstream of *mltG* were expressed in *trans* to ameliorate polar effects of *mltG* disruption (*Figure 1A*, *Figure 1—figure supplement 1A and B*). The resulting Δ6 LTG *mltG::kan* mutant was viable in LB but failed to grow in low-salt LB (LB without added NaCl, hereafter designated LB0N) (*Figure 1—figure supplement 1C*), which could explain why previous attempts to inactivate *mltG* in this background using SacB-based allelic exchange (requiring selection on LB0N + sucrose) were unsuccessful (*Weaver et al., 2019*). Consistent with these data, we were then able to generate a clean, viable Δ7 LTG mutant (*rlpA⁺ mltG::stop ΔmltABCDF Δslt70*) using an MqsR toxin-based allelic exchange system (*Lazarus et al., 2019*). The Δ7 mutant sacculus contained only a tenth of the anhMurNAc residues observed in the wild-type sacculus (*Figure 1—figure supplement 1D*, *Supplementary file 1*), suggesting that there is significantly reduced LTG activity in the Δ7 LTG mutant. Additionally, by placing the native copy of *rlpA* under an arabinose-inducible promoter, we were able to conditionally deplete RlpA by growing these strains in the absence of arabinose to observe the effects of LTG insufficiency (*Figure 1B*, *Figure 1—figure supplement 2*). Compared to Δ6 LTG (*Weaver et al., 2019*), RlpA depletion was more severe in the Δ7 background, both by morphology and plating efficiency, consistent with Δ7 exhibiting more limited LTG activity than Δ6.

Importantly, we confirmed that growth of Δ7 LTG depended on RlpA LTG activity, as RlpA^D145A, a predicted active site mutant (*Jorgenson et al., 2014*; *Figure 1—figure supplement 3A and D*), was unable to promote growth despite being stably produced (*Figure 1—figure supplement 4*) and maintaining septal recruitment as indicated by the septal localization of an RlpA^D145A-mCherry fusion (*Figure 1C*, *Figure 1—figure supplement 4*). Conversely, we found that a truncated RlpA mutant lacking the conserved SPOR domain (which is essential for septal localization in *Pseudomonas*

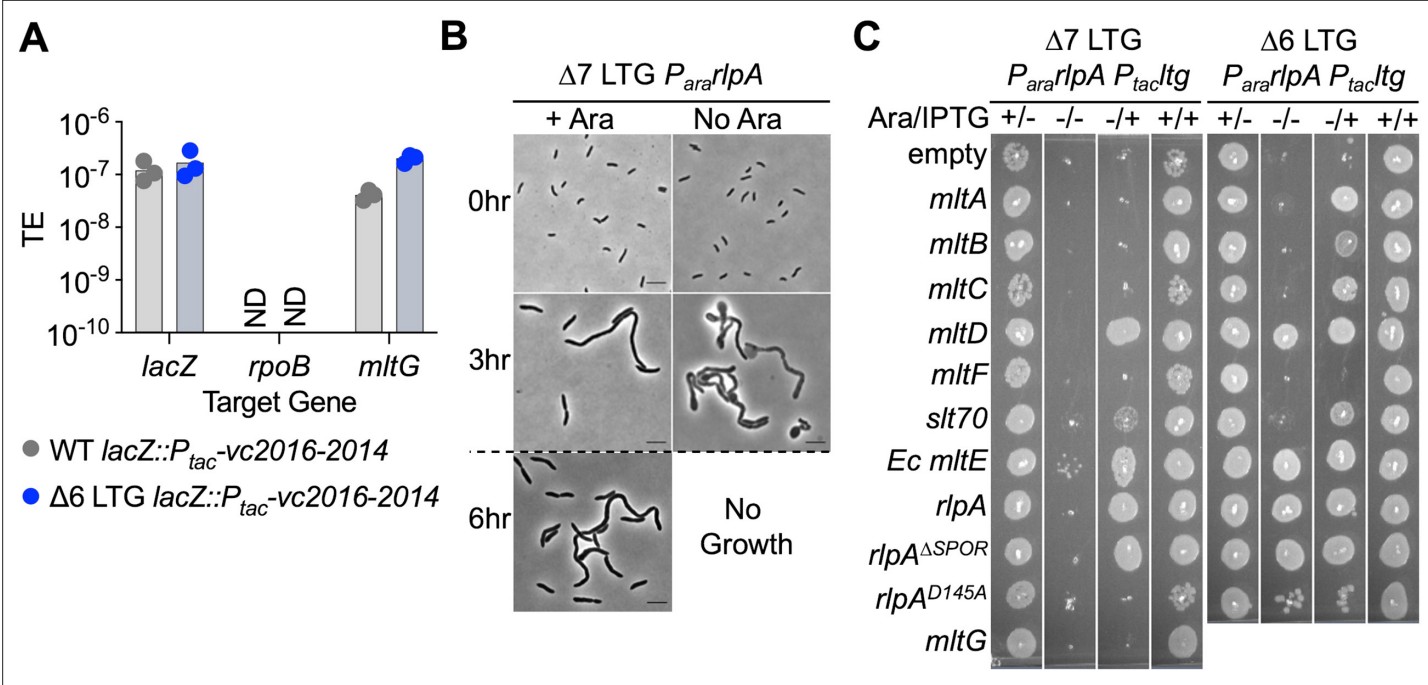

**Figure 1.** A single lytic transglycosylase (LTG) is necessary and sufficient for *V. cholerae* growth and envelope homeostasis. (**A**) *Trans* expression of DNA synthesis genes *vc2016-2014* permitted pAM299 disruption of native *mltG* locus. *lacZ* and *rpoB* were targeted as positive and negative controls for disruption, respectively. TE, transformation efficiency; ND, below limit of detection. Three biological replicates are shown. (**B**) RlpA was depleted from the WT, Δ6, and Δ7 LTG backgrounds by placing its native promoter under control of arabinose induction and growing from a $10^{-3}$ overnight culture dilution into 5 mL LB ± 0.4% arabinose (ara) at 37°C with shaking for 3 hr, back-diluting $10^{-3}$ into fresh media, and incubating for another 3 hr. Cells were imaged on LB agarose pads. Scale bars = 5 µm. Dotted line indicates $10^{-3}$ back-dilution. (**C**) Arabinose-dependent RlpA depletion in Δ6 and Δ7 LTG backgrounds was rescued with isopropyl-β-D-1-thiolgalactopyranoside (IPTG)-inducible LTGs by growing cultures in LB ± ara (0.4%) and ±IPTG (200 µM) in 96-well plates at 37°C without shaking for 3 hr, back-diluting $10^{-3}$ into fresh media, incubating another 3 hr, and spotting directly onto the same media + kan50. Plates were incubated at 30°C for 24 hr before imaging. Complete plating efficiencies associated with panels (**B**) and (**C**) can be found in *Figure 1—figure supplement 2* and *Figure 1—figure supplement 4*, respectively. Images are representative of three biological replicates.

The online version of this article includes the following source data and figure supplement(s) for figure 1:

**Source data 1.** Raw and uncropped mCherry Western blots.

**Figure supplement 1.** MltG is dispensable for growth of Δ6 lytic transglycosylase (LTG).

**Figure supplement 2.** RlpA depletion in the Δ6 and Δ7 lytic transglycosylase (LTG) backgrounds.

**Figure supplement 3.** RlpA active site architecture and localization.

**Figure supplement 4.** Lytic transglycosylases (LTGs) have variable ability to sustain growth.

---

*aeruginosa*; *Jorgenson et al., 2014*) no longer localized to the division septum in *V. cholerae* as an mCherry fusion (*Figure 1—figure supplement 3C*) yet still fully complemented native RlpA depletion in both Δ6 and Δ7 (*Figure 1C*, *Figure 1—figure supplement 4*), suggesting that RlpA LTG activity, but not septal localization, is essential in these backgrounds. Taken together, these results demonstrate that at least during growth in standard laboratory conditions, *V. cholerae* requires at minimum one active LTG of the eight currently annotated in its genome.

## Only a subset of LTGs can independently fulfill all essential LTG functions

Since RlpA LTG activity, but not septal localization, was essential for Δ6 and Δ7 growth, we asked whether growth required specialized LTG function or just PG cleavage function in general. To test this, we assessed the ability of other LTGs to complement Δ6 and Δ7 (*Figure 1C*, *Figure 1—figure supplement 4*). Only two native LTGs, MltD and Slt70, could fully (MltD) or partially (Slt70) substitute for RlpA in the Δ7 LTG background, demonstrating that RlpA, MltD, and (to a lesser degree) Slt70 are the only *V. cholerae* LTGs capable of fulfilling all required LTG roles. Additional LTGs, MltA, MltB, and MltC,

were able to rescue RlpA depletion in the Δ6 LTG background (where MltG is present), but not the Δ7 LTG background (where MltG is absent). These observations suggest that LTGs perform at least two separable essential functions for viability, where MltG can perform one function but requires MltA, MltB, or MltC to perform another, and vice versa, as none of these LTGs are independently capable of supporting growth. Intriguingly, the *E. coli* LTG MltE, for which *V. cholerae* has no known homologue, was also capable of fully supporting *V. cholerae* growth as the sole functioning LTG, suggesting that LTG essential functions may not depend on specific protein-protein interactions. Collectively, these data suggest that LTGs are partially redundant, but some LTGs also exhibit varying degrees of functional specificity.

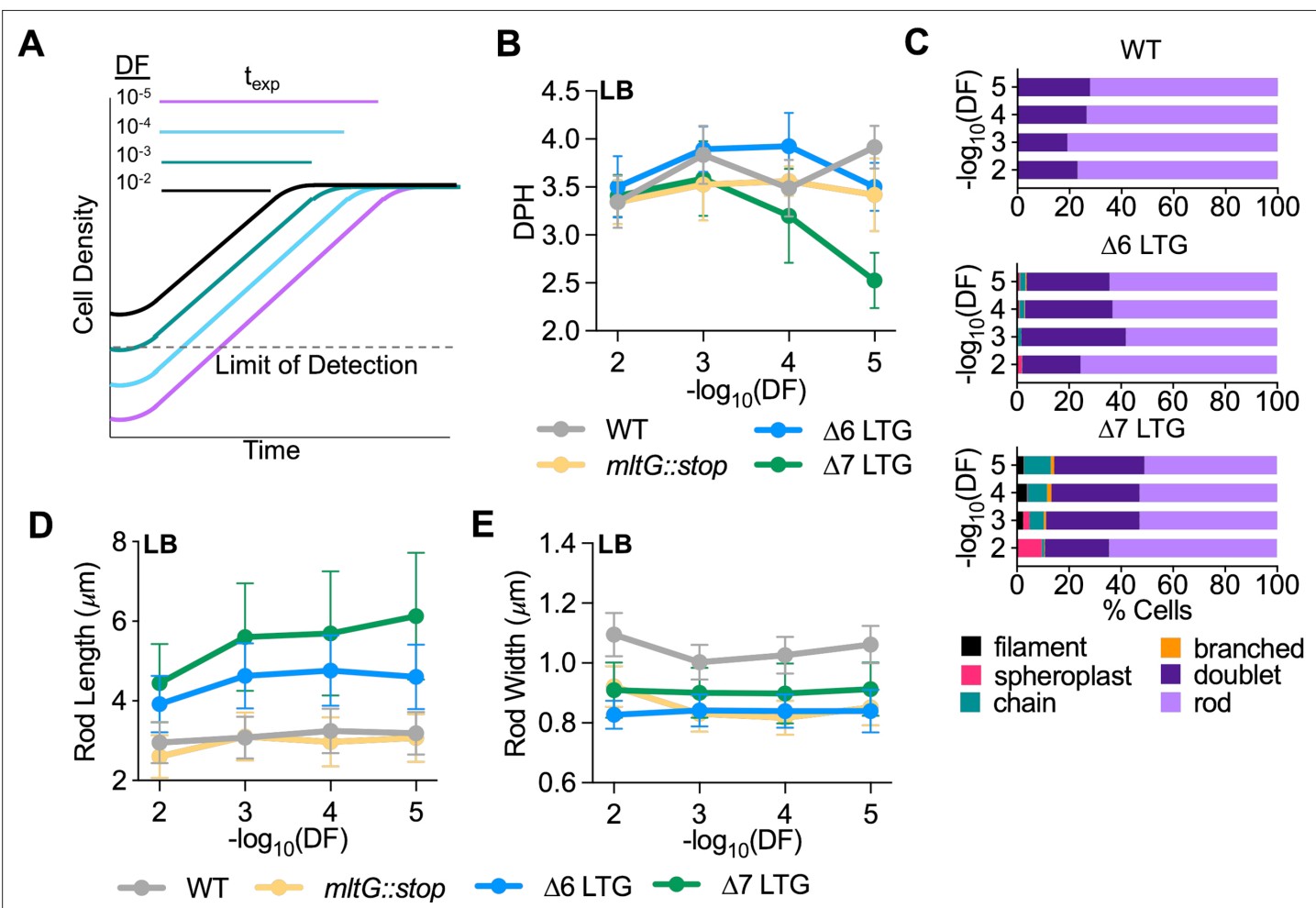

**Figure 2.** Lytic transglycosylase (LTG) insufficiency causes cumulative growth and morphology defects. (**A**) Schema describing relationship between dilution factor (DF) of saturated cultures into fresh media and time spent in exponential growth ($t_{exp}$). (**B**) Mass doubling times (doublings per hour [DPH]) from growth curves performed in LB inoculated with 10-fold serial dilutions of saturated overnight cultures. Values were calculated from growth curves shown in *Figure 2—figure supplement 2*. Error bars represent standard deviation of the mean, n ≥ 3. (**C**) Relative abundance of cell morphologies from cultures at $OD_{600}$ 0.3 from panel (**C**). n > 500 cells. Definition criteria and images are shown in *Figure 2—figure supplement 1*. Mean length (**D**) and width (**E**) as a function of DF of rod cells from panel (**C**). n > 500 rods.

The online version of this article includes the following figure supplement(s) for figure 2:

**Figure supplement 1.** Definitions of morphology defect categories.

**Figure supplement 2.** Quantitative growth and morphology of lytic transglycosylase (LTG)-deficient mutants in LB during extended exponential phase.

**Figure supplement 3.** Morphology of lytic transglycosylase (LTG)-deficient mutants in LB is growth-phase dependent.

## LTGs are required during vegetative growth

We were surprised that our strains lacking major LTG activity (even Δ7) were viable. We thus sought to characterize growth and morphology of these backgrounds in more detail. While the Δ6 and Δ7 mutants grew at wild-type growth rates in LB medium when cultures were started from a 100-fold dilution of saturated overnight cultures, we noticed a dilution-dependent exacerbation of growth and morphology defects (*Figure 2*, *Figure 2—figure supplement 1*, *Figure 2—figure supplement 2*). When Δ7 LTG cultures were highly diluted from overnight cultures to increase the number of generations spent in exponential growth, we started to observe a marked growth defect (*Figure 2A and B*, *Figure 2—figure supplement 2A*). Conversely, the *mltG::stop* single mutant and Δ6 LTG mutant grew at wild-type growth rates independent of initial dilution factor (*Figure 2B*, *Figure 2—figure supplement 2A*). These data suggest an essential or near-essential role for MltG during sustained exponential growth, albeit only when other LTGs are also inactivated. We also observed diverse aberrant morphologies within the LTG-deficient mutants (but not a Δ*mltG*::stop single mutant) under these conditions (*Figure 2C*, *Figure 2—figure supplement 1*). Interestingly, despite lacking most members of an entire class of PG enzymes, Δ6 and Δ7 only exhibited only mild defects in length and width homeostasis (*Figure 2D and E*, *Figure 2—figure supplement 2*), which were only dilution factor-dependent for cell length (suggesting a cumulative division defect). When we followed a time course after back-dilution from stationary phase, we observed morphological defects accumulating in exponential phase and then largely disappearing in stationary phase (*Figure 2—figure supplement 3*) in both Δ6 and Δ7. Collectively, these observations suggest that LTG-deficient mutants suffer from cumulative damage during exponential growth, which is partially alleviated in stationary phase.

## LTG activity is required for survival in hypo-osmotic conditions

Our results detailed above suggested that the majority of LTGs are dispensable for growth, catalyzing renewed interest in the question of what their physiological roles are. To dissect potential roles for LTGs in cell envelope integrity maintenance, we subjected the Δ6 and Δ7 mutants to growth in low osmolarity medium. Similar to the morphology and growth defects in LB, both mutants were sensitive to low-salt conditions in a dilution-dependent manner (*Figure 3A*, *Figure 3—figure supplement 1*). Interestingly, the Δ7 LTG mutant grew at wild-type rate during the initial growth period before a rapid decrease in $OD_{600}$ (indicative of lysis), suggesting that it is not the initial shock of changing osmotic conditions that kills this mutant, but rather some cumulative damage during growth in low-salt conditions (*Figure 3*, *Figure 3—figure supplement 1A*). Importantly, this rules out a simple cell envelope defect as a cause of LTG-deficient mutant osmosensitivity. We then sought to dissect the contributions of individual LTGs to salt sensitivity. Other than RlpA, which has an established role in LB0N growth (*Jorgenson et al., 2014*; *Weaver et al., 2019*) and is still present in the Δ6 and Δ7 mutants, no other single LTG mutant exhibited a strong growth defect in LB0N (*Figure 3—figure supplement 2*), all further suggesting that this defect is an accumulative function of collective LTG insufficiency. Several LTGs (MltA, MltB, Slt70, and EcMltE) were able to rescue Δ6 LTG growth in LB0N at higher dilutions (*Figure 3—figure supplement 2*), but none could rescue Δ7 growth in LB0N except MltG, which only partially restored growth to Δ6 mutant levels (*Figure 3—figure supplement 2*). This suggests that no single LTG can fully support growth in LB0N, and that MltG, despite not exhibiting a defect as a single mutant, is particularly important in this environment when other LTGs are inactivated. Intriguingly, overexpression of *mltF* was toxic for both Δ6 (LB0N only) and Δ7 mutants (LB) (*Figure 3—figure supplement 2C and D*), preventing us from assessing its potential for LTG-deficient mutant complementation.

## LTG mutants are hypersensitive to accumulation of periplasmic polysaccharides

To determine the reason for ΔLTG defects in low-salt media, we took advantage of the spontaneous appearance of suppressors arising during Δ6 LTG growth in LB0N (*Figure 3B*, *Figure 3—figure supplement 1B*). Whole-genome sequencing of three stable suppressors identified three unique mutations: a deletion in *ptsH* (*vc0966*), a frameshift mutation in *opgH* (*vc1287*), and a deletion mutation affecting both *opgH* and a downstream gene (*vc1286*) (*Figure 3C*). *ptsH* encodes HPr, a key regulator of sugar import via the phosphoenolpyruvate-carbohydrate phosphotransferase system (PTS) (*Deutscher et al., 2014*). OpgH is critical to the synthesis of periplasmic glucans (OPGs), which accumulate under

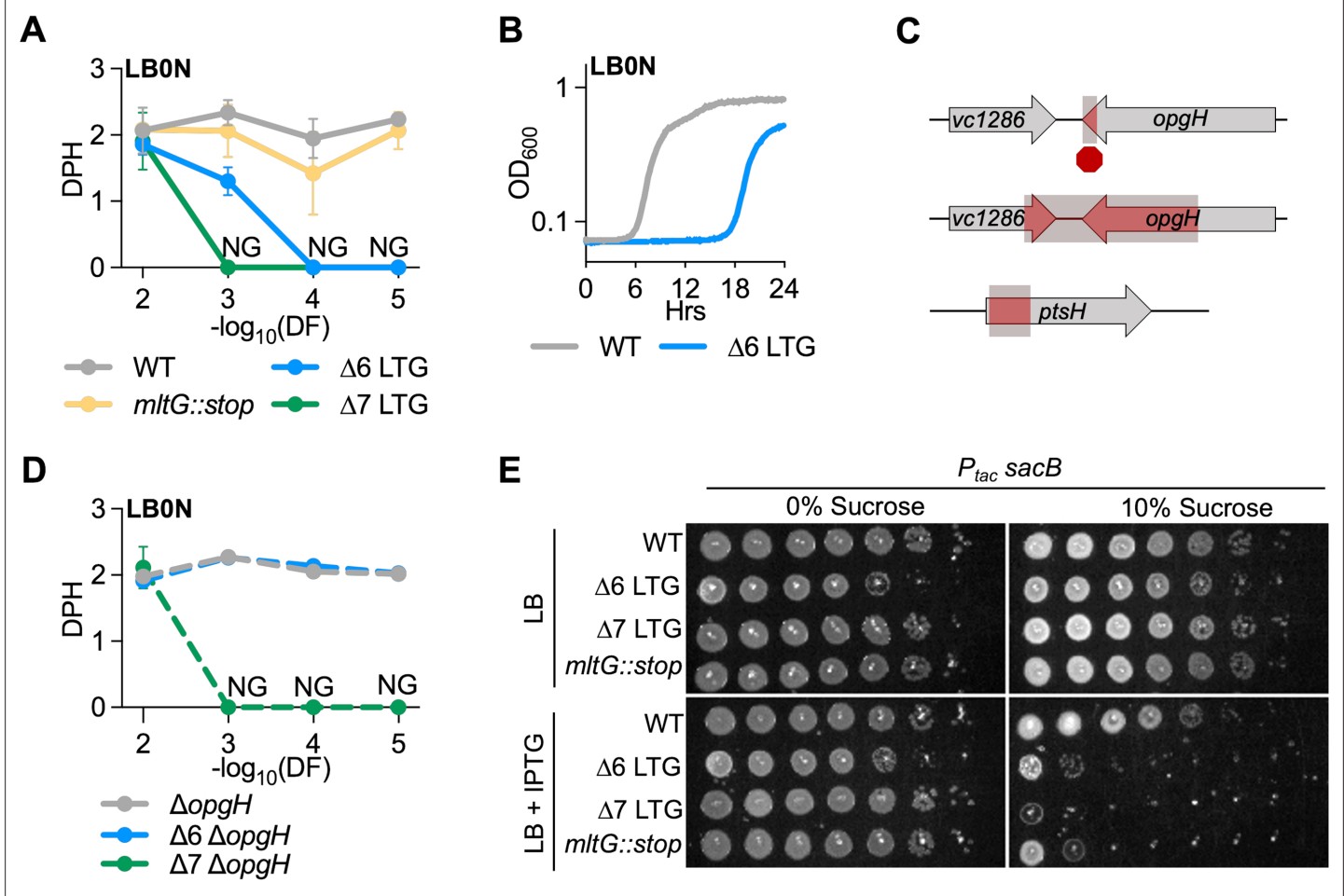

**Figure 3.** Lytic transglycosylase (LTG) mutants are hypersensitive to low osmolarity and accumulation of periplasmic polymers. (**A**) Mass doublings (doublings per hour [DPH]) upon increasing dilutions (dilution factor [DF]) during growth in LB0N. Values were calculated from growth curves in *Figure 3—figure supplement 1*. Error bars represent standard deviation of the mean, n ≥ 3. NG, no growth. (**B**) Representative growth curve in low-salt LB (LB0N) showing late-growing spontaneous suppressor in Δ6 LTG. (**C**) Whole-genome sequencing of Δ6 LB0N suppressor mutations identifies a premature stop, resulting in a 6% 3' truncation of *opgH;* a deletion resulting in a 56% 3' truncation of *opgH* and 36% 3' truncation of *vc1286;* and a deletion of the 5' end of *ptsH.* (**D**) Validation of low osmolarity growth defect suppression by an *opgH* mutation. Shown are mass doublings (DPH) upon increasing dilutions (DF) during growth in LB0N. Values were calculated from growth curves shown in *Figure 3—figure supplement 1*. (**E**) Saturated overnight cultures harboring isopropyl-β-D-1-thiolgalactopyranoside (IPTG)-inducible *sacB* were 10-fold serially diluted and plated on LB + kan50 ± 200 μM IPTG ± 10% sucrose, incubated at 30°C, and imaged 24 hr before. Representative of three biological replicates. Empty vector and LB0N controls are shown in *Figure 3—figure supplement 4*.

The online version of this article includes the following source data and figure supplement(s) for figure 3:

**Source data 1.** Raw growth curve data for single lytic transglycosylase (LTG) mutants and LTG complementation in LTG-deficient mutants.

**Figure supplement 1.** Suppression of growth defects of ΔLTG mutants in low-salt LB.

**Figure supplement 2.** Single LTG contributions to growth in LB and low salt LB.

**Figure supplement 3.** Periplasmic glucans do not contribute to division defects in ΔLTG mutants.

**Figure supplement 4.** SacB expression is toxic in ΔLTG mutants in a sucrose-dependent manner.

conditions of low osmolarity and have been implicated in a variety of cell functions, including steady-state maintenance of osmolarity in the periplasm (*Bontemps-Gallo et al., 2017*). We were particularly intrigued by the role of periplasmic glucans and validated restoration of Δ6 LTG LB0N growth using clean deletion mutants. Inactivation of *opgH* completely restored wild-type growth of the Δ6 LTG mutant in LB0N, independent of initial dilution factor (*Figure 3D*, *Figure 3—figure supplement 1E*). In contrast, inactivation of *opgH* was only able to restore growth yield of the Δ7 LTG mutant in LB0N from a $10^{-2}$ inoculum (*Figure 3D*, *Figure 3—figure supplement 1E*), but not from greater dilutions,

nor from a $10^{-5}$ inoculum in LB (*Figure 3—figure supplement 1C and D*), indicating that the Δ7 LTG mutant has additional exponential growth-dependent defects unique from the Δ6 LTG mutant. Interestingly, OpgH orthologues in *E. coli* and *Yersinia pseudotuberculosis* have also been ascribed a moonlighting function, tying carbon availability with cell length by inhibiting FtsZ filamentation when UDP-glucose levels are high in the cell (*Hill et al., 2013*; *Quintard et al., 2015*). The elongated cell phenotype of the Δ6 LTG and Δ7 LTG mutants could in principle be indicative of a similar activity by *V. cholerae* OpgH; however, deleting *opgH* from the Δ6 LTG mutant did not restore wild-type cell length in Δ6 (*Figure 3—figure supplement 3*). Additionally, alignment of *V. cholerae* OpgH with *E. coli* and *Y. pseudotuberculosis* suggests that it lacks much of the N-terminal domain that was shown to interact with FtsZ in *E. coli* (*Figure 3—figure supplement 3*; *Hill et al., 2013*).

We consequently considered a model where accumulation of periplasmic glucans is detrimental to ΔLTG mutants, for example, via periplasmic crowding or an increase in periplasmic osmolarity. To test this model, we used an unrelated system to increase periplasmic crowding. The *Bacillus subtilis sacB* gene product, which is secreted and functions extracellularly (*Pereira et al., 2001*), isomerizes and polymerizes sucrose monomers into levan molecules up to several kD (*Tanaka et al., 1980*), molecules much too large to escape the Gram-negative periplasm through outer membrane porins. This is often exploited in allelic exchange methods for mutant generation as a means of counterselection (*Steinmetz et al., 1983*; *Gay et al., 1983*; *Blomfield et al., 1991*), where, for example, WT *V. cholerae* is sensitive to *sacB* expression in LB0N + sucrose. We hypothesized that the ΔLTG mutants might be hypersensitive to periplasmic levan synthesis even in standard LB salt conditions (1% W/V). To test this, we engineered strains overexpressing *sacB*. Consistent with the use of *sacB* as a counterselection method under low-salt conditions, the wild-type is sensitive to *sacB* induction on LB0N (*Figure 3—figure supplement 4*), but not LB. In contrast, we observed a *sacB*-dependent plating defect on sucrose in the Δ6 and Δ7 LTG mutant on LB (*Figure 3E*, *Figure 3—figure supplement 4*). Interestingly, the *mltG::stop* mutant also exhibited hypersensitivity to SacB activity (*Figure 3E*, *Figure 3—figure supplement 4*), suggesting that accumulating MltG substrate is a particularly strong direct or indirect contributor to this phenotype. Collectively, our data suggest that ΔLTG mutants suffer from either hyperosmotic stress or excessive molecular crowding in the periplasm, which can be exacerbated through the induction of long-chain polysaccharides.

## PG transglycosylase activity causes Δ6 LTG mutant periplasmic stress

Based on our model that accumulation of polysaccharides in the periplasm is toxic, we predicted that accumulation of uncrosslinked PG strands caused by uninterrupted GT activity during TP inhibition, for example, by β-lactams (*Cho et al., 2014*) (a process termed 'futile cycling' when coupled with LTG-mediated degradation), should exacerbate LTG-deficient mutant sickness, and this has indeed been shown with Slt70 in *E. coli* (*Cho et al., 2014*). To test this hypothesis, we assessed susceptibility of ΔLTG mutants to cell wall-acting antibiotics with varying ability to induce futile cycling. In a disk diffusion assay, the Δ6 and Δ7 LTG mutants were hypersensitive to inducers of futile cycling, that is, general PBP inhibition by Penicillin G (PenG) as well as to inhibition of specific PBPs including PBP3 (aztreonam), PBP2 (mecillinam), and PBP1b (cefsulodin) (*Figure 4A*). Conversely, both Δ6 and Δ7 mutants exhibited wild-type sensitivity to moenomycin and fosfomycin, both of which inhibit cell wall synthesis without inducing futile cycling. Thus, β-lactam sensitivity of LTG-deficient mutants is not necessarily tied to simple inhibition of cell wall synthesis, but potentially also to periplasmic crowding due to the accumulation of uncrosslinked PG strands. Curiously, the Δ7 LTG mutant was hypersensitive to MreB inhibition by MP265. This suggests that in the absence of other LTGs MltG contributes to survival upon Rod system insufficiency through an unknown mechanism.

To further dissect PG strand accumulation under these conditions, we visualized PG synthesis and turnover during antibiotic treatment using the cell wall label BADA (*Hsu et al., 2017*; *Kuru et al., 2012*). The Δ6 and Δ7 LTG mutants were all readily labeled when grown with BADA (*Figure 4—figure supplement 1*, *Figure 4—figure supplement 2*) prior to antibiotic addition. Upon treatment with cell wall-targeting antibiotics, WT *V. cholerae* degrades its structural, rod-shaped sacculus to ultimately yield stable, cell wall-deficient spheroplasts (*Dörr et al., 2015*; *Weaver et al., 2018*). After up to 3 hr of treatment with PenG (100 µg/mL, 20× MIC) and continued incubation with BADA, the Δ6 and Δ7 LTG mutant spheroplasts accumulated strong periplasmic BADA signal compared to the wild-type spheroplasts in which cell wall material is presumably completely degraded and removed from the

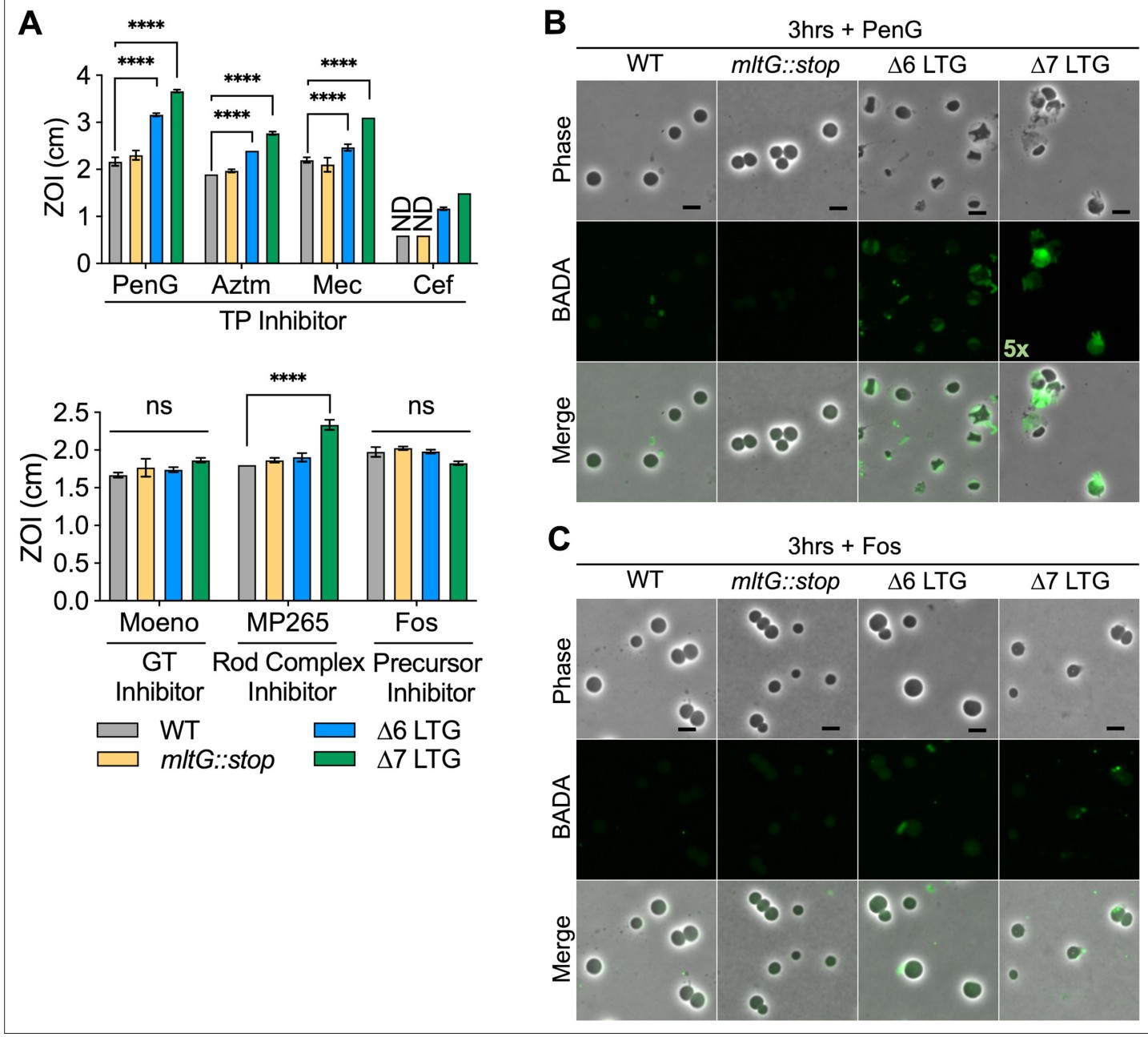

**Figure 4.** Lytic transglycosylase (LTG) mutants are hypersensitive to antibiotics promoting periplasmic peptidoglycan (PG) accumulation. (**A**) Sensitivity to Penicillin G (PenG), aztreonam (AZTM), mecillinam (Mec), cefsulodin (Cef), moenomycin (Moeno), MP265, and fosfomycin (Fos) measured as zone of inhibition (ZOI) in a disk diffusion assay. ND, no ZOI around disk. Error bars, standard deviation. Significance determined by one-way ANOVA. ns, $p > 0.05$; *$p < 0.05$; **$p < 0.01$; ***$p < 0.001$; ****$p < 0.0001$. Overnight cultures were diluted 1:100 into LB + BADA (100 µM) and grown at 37°C to $OD_{600}$ 0.5 before addition of (**B**) PenG (100 µg/mL) or (**C**) Fos (500 µg/mL). Resulting spheroplasts were washed and imaged after 3 hr of antibiotic exposure. Fluorescence was normalized to the same intensity threshold for visual comparison except where indicated (exceptionally bright samples were normalized to a higher-intensity threshold denoted by the multiplier). Representative of two biological replicates. Scale bar = 5 µm.

The online version of this article includes the following figure supplement(s) for figure 4:

**Figure supplement 1.** Periplasmic cell wall accumulation in response to Penicillin G (PenG).

**Figure supplement 2.** Lack of periplasmic cell wall accumulation in response to fosfomycin.

**Figure supplement 3.** Defects of lytic transglycosylase (LTG)-deficient mutants are independent of peptidoglycan (PG) recycling.

periplasm (*Figure 4B*, *Figure 4—figure supplement 1*). Meanwhile, in the Δ6 and Δ7 LTG mutants, uncrosslinked PG likely accumulates due to aberrant GT activity. Alternatively, degraded PG from the sacculus may be sufficiently retained in the periplasm to be additionally labeled with BADA by L,D-transpeptidases (*Kuru et al., 2017*) in the Δ6 and Δ7 LTG but not in the WT. However, inhibition of cell wall synthesis upstream of GT activity using fosfomycin (500 µg/mL) (which is expected to result in sacculus degradation without induction of futile cycling) did not result in pronounced BADA accumulation in the periplasm, suggesting that the increasing BADA signal observed during penicillin treatment does not solely reflect retention of labeled debris from the degraded sacculus, but indeed is the result of ongoing cell wall synthesis (*Figure 4B*, *Figure 4—figure supplement 2*). The diffuse nature of the BADA signal additionally suggests that the PG in the Δ6 and Δ7 spheroplasts is solubilized debris trapped in the periplasm, as opposed to a thin, intact layer of peripheral PG. At earlier stages of penicillin treatment, it also appeared that the Δ6 and Δ7 LTG mutants were slower to degrade their rod-shaped poles (*Figure 4—figure supplement 1*).

## PG recycling is not required during periplasmic stress

The roles for LTGs in PG recycling are well characterized. LTGs are required to digest longer glycan strands down to single disaccharide subunits that can be imported into the cytoplasm via AmpG, a permease that selectively recognizes the anhMurNAc residue generated by LTG activity. Mutants lacking AmpG have been demonstrated to accumulate extracellular monomeric disaccharide LTG-turnover products (*Hernández et al., 2020*). We therefore asked whether a lack of recycling could account for some of our key phenotypes. However, an Δ*ampG* mutant exhibited wild-type behavior for growth in LB0N, sacB overexpression, β-lactam resistance, and BADA staining after PenG exposure (*Figure 4—figure supplement 3*), demonstrating that lack of PG recycling does not promote the LTG-deficient phenotypes observed here.

## LTG insufficiency results in periplasmic PG strand accumulation

Why are LTG mutants sensitive to periplasmic accumulation of polymers? Accumulating PG debris released during PG synthesis could, in principle, increase periplasmic osmolarity and/or crowding, explaining defects associated with low-salt conditions and periplasmic polysaccharide accumulation. We therefore sought to quantify soluble, periplasmic PG debris within crude cell lysates (excluding PG released into the growth medium). Canonical PG architecture analysis relies on SDS-boiled sacculi isolated by ultra-centrifugation, which permits sacculus characterization (*Figure 1—figure supplement 1D*, *Supplementary file 1*), but ignores solubilized, uncrosslinked (freely diffusing) PG fragments associated with PG turnover processes such as those potentially mediated by LTGs. We thus analyzed fragments that remained soluble after sedimentation of purified sacculi, that is, PG material that freely accumulates within the periplasm (or inside the cell) but is not attached to the cell wall. Since entire periplasmic PG strands cannot be easily resolved using subsequent LC-MS analysis, we also digested the soluble PG fraction with muramidase (to generate smaller monomers suitable for LC-MS detection) and then compared muramidase-treated vs. untreated traces to determine soluble PG architecture (*Figure 5A*). We first analyzed soluble products of LTG activity (*Figure 5BC*, *Supplementary file 2*). AnhMurNAc-tetrapeptide (M4N) was abundant in the wild-type during exponential phase, and predictably, M4N was significantly depleted in the Δ7 LTG-soluble muropeptide profile. Surprisingly, M4N was enriched in the *mltG::kan* and Δ6 LTG mutants, suggesting that the absence of some LTGs might cause upregulated activity of others.

Importantly, MurNAc-tetrapeptide species without anhMurNAc (M4) were detectable in the WT during exponential phase and significantly enriched in the Δ6/Δ7 LTG mutants (*Figure 5D*, *Supplementary file 2*). By comparing the muramidase-treated vs. untreated samples, we can infer the native state of these M4 and M4N (*Figure 5A, B, and D*), that is, determine whether these species occur predominantly as monomers or as parts of PG polymers in vivo. Both species were significantly depleted in the muramidase-untreated samples, suggesting that in vivo they are part of polymeric, uncrosslinked PG strands. It is important to note that M4 monomers are not intermediates of cell wall synthesis (which proceeds from a tripeptide directly to pentapeptide due to addition of preformed D-Ala-D-Ala dipeptide) (*Barreteau et al., 2008*). This, in conjunction with a strong muramidase-treatment-dependence of the abundance of M4, excludes a cytoplasmic origin of this species. It is also interesting to note that, unlike the WT, *mltG::kan*, and even Δ6 LTG mutant, the Δ7 LTG appears

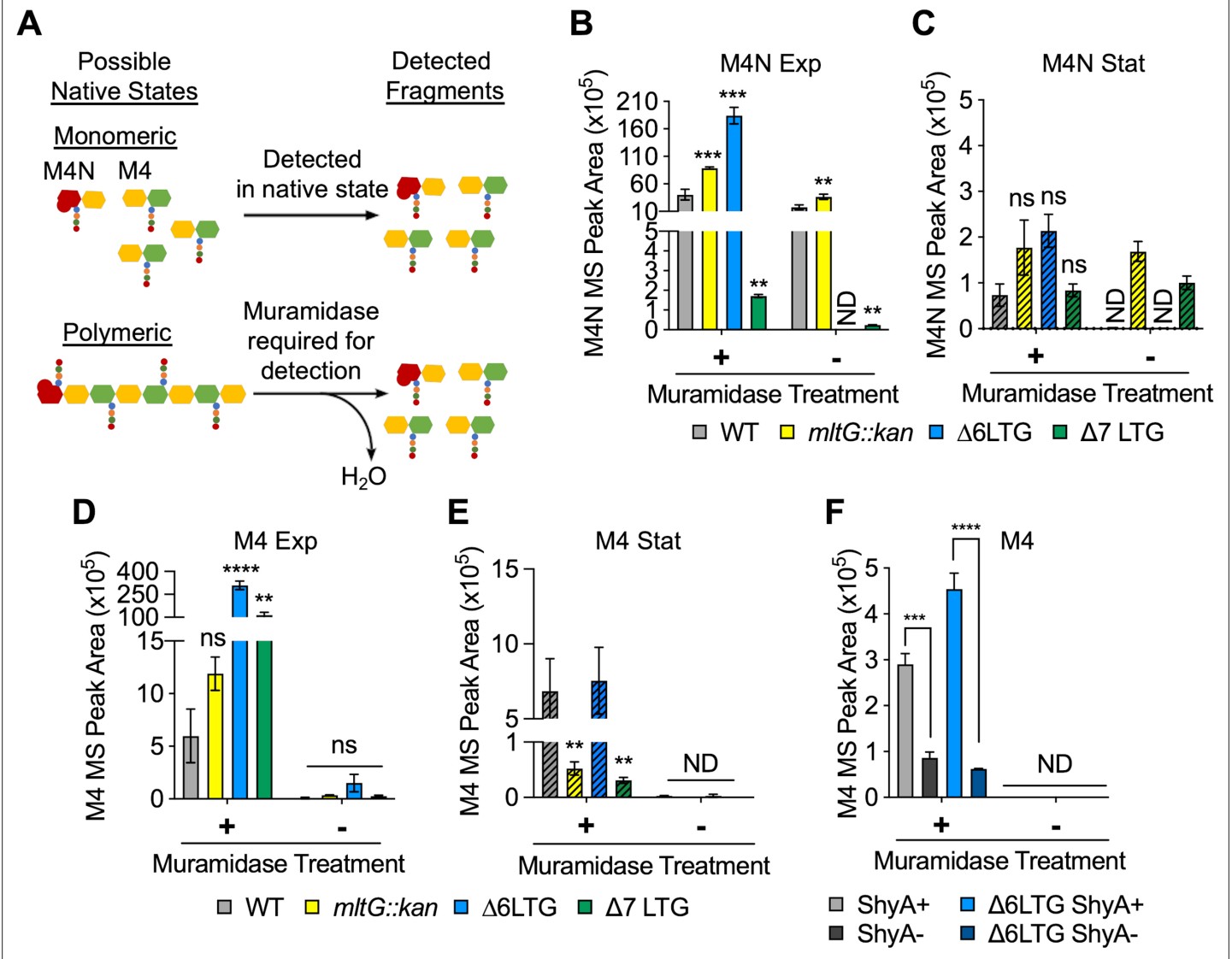

**Figure 5.** Periplasmic uncrosslinked peptidoglycan (PG) strands accumulate in an endopeptidase-dependent manner during normal growth. (**A**) Schema describing muramidase treatment dependence of detection of monomeric or polymeric PG fragments. M4N, anhMurNAc-tetrapeptide; M4, reduced MurNAc-tetrapeptide. (**B–E**) Overnight cultures of. WT and ΔLTG mutants were diluted 1:100 into LB, grown at 37°C, and harvested at $OD_{600}$ 0.3 (Exp, solid bars) and 1.2 (Stat, striped bars) for soluble PG analysis by LC-MS. MS peak areas for M4N and M4 are shown here, and complete muropeptide profiles can be found in *Supplementary file 2*. Means compared to WT by unpaired *t*-tests, n = 3. (**F**) WT and Δ6 LTG strains harboring a single chromosomal copy of *shyA* under an isopropyl-β-D-1-thiolgalactopyranoside (IPTG)-inducible promoter were grown from $10^{-2}$ inocula for 3 hr ($OD_{600}$ ~ 1.0) in LB with (ShyA+) or without (ShyA -) 200 μM IPTG at 37°C and harvested for soluble PG analysis by LC-MS. Complete muropeptide profiles can be found in *Supplementary file 3*. Means compared by unpaired *t*-test, n = 3. All error bars = standard deviation. ns, p>0.01; **p<0.01; ***p<0.001; ****p<0.0001. ND, not detected in all replicates.

The online version of this article includes the following figure supplement(s) for figure 5:

**Figure supplement 1.** Endopeptidase depletion phenotypes in lytic transglycosylase (LTG)-deficient mutants.

to have a disproportionately large pool of polymeric M4 species (muramidase-dependent) compared to M4N species, suggesting that either the soluble PG strands in the Δ7 LTG mutant are extremely long or perhaps do not ubiquitously terminate in anhMurNAc residues. The abundance of soluble PG strands decreased in stationary phase (*Figure 5CE*, *Supplementary file 2*), which suggests that in the LTG-deficient mutants these strands are somehow cleared by either RlpA, an unrecognized LTG, or a cryptic PG hydrolase. Reduction of these uncrosslinked strands in stationary phase is consistent with

the alleviation of morphology defects observed as cells exit exponential growth (*Figure 2—figure supplement 3*).

We next asked how these polymeric, uncrosslinked PG strands might be generated. Since EPs have been suggested to be essential for sacculus expansion during cell elongation, we hypothesized that these soluble strands may reflect EP activity, which would imply that EPs do not simply relax PG crosslinking (as commonly assumed), but also might excise entire strands that accumulate – at least transiently – in the periplasm. To test this hypothesis, we depleted the major housekeeping EP ShyA (*Dörr et al., 2013*) from a Δ6 LTG mutant and found that the most abundant monomer species, M4, was reduced in a ShyA-dependent manner in both WT and Δ6 LTG mutants (*Figure 5F*, *Supplementary file 3*). This strongly suggests that ShyA produces the majority of the uncrosslinked strands in the periplasm. Curiously, depletion of ShyA in the Δ6 LTG background resulted in large, irregular cells (*Figure 5—figure supplement 1*), suggesting a cell envelope defect upon EP insufficiency. Additionally, ShyA was required for Δ6 LTG mutant colony formation on LB plates, but Δ6 LTG liquid cultures depleted for ShyA remained viable and could be rescued on plates restoring *shyA* expression (*Figure 5—figure supplement 1*). Altogether, the data suggest that EPs excise entire PG strands that accumulate in the periplasm until they are later cleared by LTGs.

## Discussion

Despite decades of work and a renewed research focus in the last few years, we still lack a fundamental understanding of how bacteria harness both PG synthesis and constant degradation to build and maintain an essential and dynamic wall structure that is able to withstand a high internal turgor pressure. Even the basic physiological function of many cell wall cleavage enzymes ('autolysins') remains unknown. The exact contributions of LTGs to cell growth, for example, have remained elusive, mainly due to the high level of apparent redundancy of these enzymes, which hampered classical genotype-phenotype association analyses. While circumstantial evidence abounds, no collective physiological characterization of essential LTG function has been conducted in any organism, and even cases of demonstrated synthetic lethal relationships are rare and rely on indirect evidence (inability to delete LTG genes) (*Heidrich et al., 2002*; *Chaput et al., 2007*; *Scheurwater and Clarke, 2008*). The *V. cholerae* Δ6 and Δ7 LTG mutants described here provide a vital platform for exploring both the collective and individual contributions of LTGs to bacterial viability.

### Conserved LTGs are not functionally equivalent

In this study, we confirm that LTG enzymatic activity is essential for growth and division. We further demonstrate that some functional redundancy exists in *V. cholerae* between diverse LTG families, as has been observed in other bacteria (*Atassi, 2017*; *Dik et al., 2017*; *Mueller et al., 2019*). Members of LTG families 1A (Slt70), 1D (MltD), and 6A (RlpA) can support growth independent of all other natively encoded LTGs from families 1B, 1E, 2A, 3A, and 5A. Certain *V. cholerae* LTGs additionally exhibit complementary genetic relationships in the absence of all other LTGs, implying there are at least two essential roles for LTGs. For example, while neither MltG, MltA, MltB, nor MltC can support growth on their own, strains expressing the MltG/ MltA, MltG/MltB, or MltG/MltC pairs are each viable. Surprisingly, and importantly, essential LTG activities may not require conserved protein-protein interactions as suggested by the viability of a *V. cholerae* mutant expressing non-native *E. coli* MltE as its sole LTG. Scheurwater and Clarke unsuccessfully but informatively attempted to inactivate *mltF* in an *E. coli* Δ6 LTG (Δ*mltABCDE* Δ*slt70*), suggesting that the essentiality of LTG activity is conserved between *E. coli* and *V. cholerae,* and that MltF and MltG may not be functionally redundant in *E. coli* (*Scheurwater and Clarke, 2008*). In contrast, *mltF* inactivation is not synthetically lethal with Δ*mltABCD* Δ*slt70* in *V. cholerae*. In fact, restoring MltF expression in the Δ6 and Δ7 LTG background is toxic, indicating that *V. cholerae* and *E. coli* exhibit different genetic relationships between homologs of the same LTGs. This is perhaps most strongly exemplified by the conservation of RlpA in both species, where *E. coli* RlpA has no detectable LTG activity (*Jorgenson et al., 2014*), yet *V. cholerae* RlpA can perform essential functions as the sole LTG. Recently, the hydrolytic enzyme DigH in *E. coli* was shown to be functionally similar to LTGs by contributing to the resolution of septal PG during daughter cell separation (*Yakhnina and Bernhardt, 2020*). The activity of DigH (of which *V. cholerae* does not possess a strong homolog) may explain why *E. coli* does not appear to require RlpA LTG activity; this LTG-like

physiological role for a hydrolytic enzyme also suggests that we may need to look beyond LTGs to fully address the roles for glycosidic bond cleavage within PG. Another recent example of glycosidases serving as functional analogues includes MpgA from *Staphylococcus aureus,* which shares enough homology with MltG to be identified bioinformatically as a YceG-family LTG but harbors a single active site mutation to impart a muramidase mechanism of PG cleavage (*Taguchi et al., 2021*).

The functional redundancy of LTGs is also likely the root of an apparent paradox. Release of new PG strands from undecaprenyl pyrophosphate should be an essential function as anchoring of PG to the inner membrane is toxic (*Suzuki et al., 2002*) and PG release is likely necessary for lipid carrier recycling. Yet MltG, the highly conserved 'PG terminase,' or more generally, 'PG release factor,' which associates with PG synthetic complexes to release new strands, is not essential. MltG can be inactivated in wild-type *V. cholerae* (and *E. coli*) without significant consequence, but our Δ7 LTG (*mltG*-null) mutant lost viability upon sustained growth in exponential phase in LB. It is therefore likely that other LTGs can partially complement for PG release activity. The dilution-dependent Δ7 LTG growth defect further suggests that either RlpA has at least partial PG release activity and/or that during brief growth in exponential phase PG release activity is nonessential. MltG essentiality (or near-essentiality) can thus perhaps only be appreciated in a multiple LTG background under conditions where PG release activity is expected to be most important. This is also evidenced by the increased sensitivity of the Δ7 LTG mutant to hypo-osmotic conditions that cannot be rescued via the same mechanisms that rescue the Δ6 LTG mutant, that is, inactivation of OPG synthesis, as well as hypersensitivity to MreB inhibitor MP265. All of these defects point to a unique, MltG-dependent, conditionally essential LTG function. While this study has focused on clearance of soluble PG debris, an activity that would perhaps only be critical in organisms with a membrane-confined periplasmic space, PG release factors analogous in function to MltG are broadly conserved even in Gram-positive organisms, additionally demonstrating a necessity for further investigation into the role of PG release in bacterial cell wall homeostasis.

## LTGs manage the periplasmic environment

By analyses of soluble PG, we discovered that even wild-type *V. cholerae* accumulates loose strands of uncrosslinked PG of periplasmic origin (detected as muramidase-dependent, soluble M4 products) and that production of these strands is dependent on the EP ShyA. PG strands appear to accumulate to detrimental levels in the Δ6 and Δ7 mutants, and we thus propose that processing soluble PG is a major collective function of LTGs. PG strand accumulation induces hypersensitivity to other polymeric

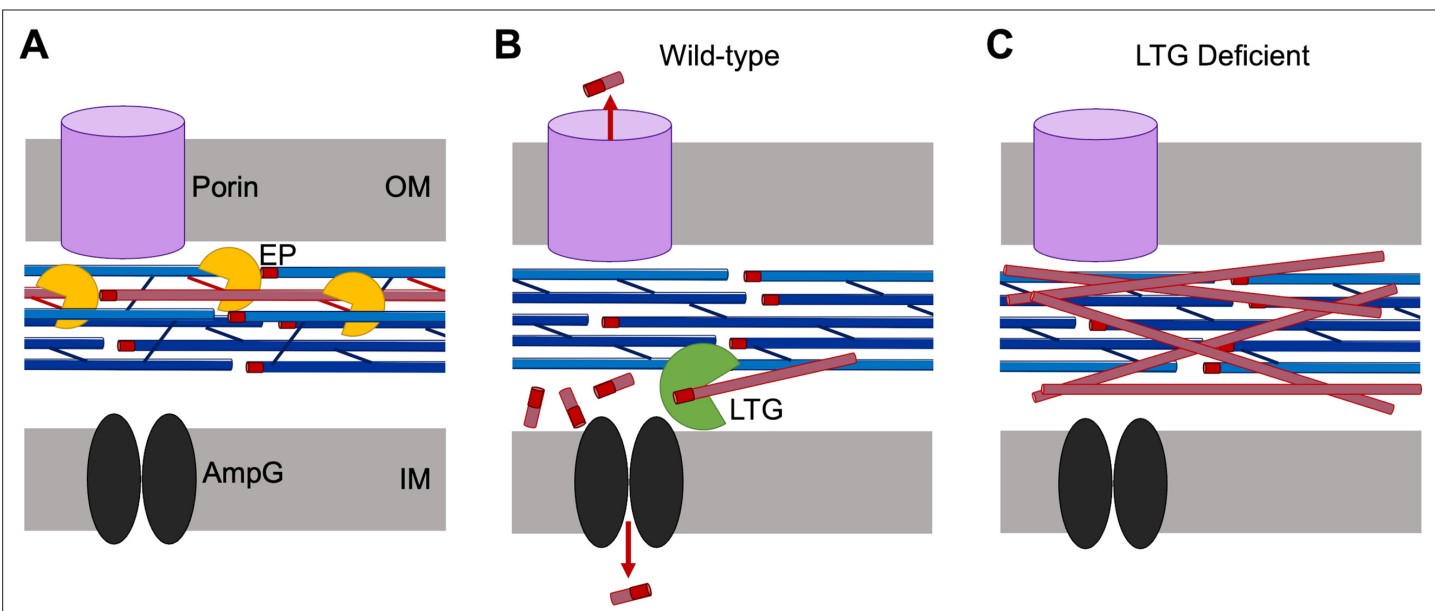

**Figure 6.** Model for lytic transglycosylase (LTG)-mediated removal of toxic peptidoglycan (PG) debris. (**A**) Endopeptidases (EPs, yellow) excise PG strands (red) from the sacculus (blue), permitting sacculus expansion. (**B**) In wild-type cells, LTGs (green) digest excised, uncrosslinked PG strands into smaller fragments that can be recycled by AmpG (black) or released through porins (violet). (**C**) In LTG-deficient cells, excised PG debris crowds the periplasm and becomes toxic.

sugars in the periplasm, such as OPGs, SacB-generated levans, and β-lactam-induced futile cycling products. Since these hypersensitivities are not shared by a ΔampG mutant, previously demonstrated to accumulate small PG monomers, this polysaccharide toxicity may arise from the fact that these large polymers (PG, OPGs, or levans) cannot diffuse out through OM porins, accumulating in a growth and/or substrate-dependent manner. This would inappropriately increase the osmolarity and/or cause excessive crowding (*Sochacki et al., 2011*) of the periplasm to interfere with normal growth-related processes, as exemplified by the mild division defect of the *V. cholerae* LTG-deficient mutants.

While periplasmic crowding can clearly occur in Gram-negative bacteria due to their diffusion-limiting OM, it is intriguing to consider implications for Gram-positives, where LTGs are likewise conserved, and the cell wall may not provide a meaningful diffusion barrier to long PG strands. We propose three possible (and testable) scenarios for the role of LTGs in Gram-positive bacteria. For example, only PG release activity might be essential in Gram-positives. It is also conceivable that LTGs are nonessential in Gram-positive bacteria. Lastly, 'periplasmic' crowding may in fact be detrimental in Gram-positive bacteria: *B. subtilis* PG, for example, is very long chained, and it is conceivable that PG may constitute a reasonable diffusion barrier against such long strands.

### Broader implications for LTGs in PG dynamics

Our data suggest that the most important roles for LTGs may not include 'making space' for new PG insertion, as has been assumed for fundamental models of PG synthesis. More likely, LTGs clear the periplasm of most PG debris generated by EPs during normal growth by facilitating either recycling or release of small fragments into the environment (*Figure 6*). As such, this collectively essential function of LTGs may not be tied to a direct influence on PG synthesis. Importantly, uncrosslinked strands are not a deviant feature unique to the LTG-deficient mutants nor β-lactam treatment (though they become more apparent under these conditions) as they were also detectable in the wild-type during normal growth. Detection of EP-dependent, uncrosslinked PG strands suggests that EP activity might be temporally separated from LTG-mediated turnover of those PG strands. This is seemingly contrary to pervasive models proposing a synchronized synthetic/autolytic complex that digests old PG while simultaneously building new PG. If such a complex exists, we propose that LTGs collectively maintain an important role complementary to it, but not inevitably as part it.

Lastly, while high accumulation of PG strands is toxic, maintaining a small pool of uncrosslinked PG material may actually serve a beneficial purpose to cells under some conditions; bacteria might thus have an adaptive reason for avoiding tightly coordinating LTG activity with synthesis. Inactivation of *sltY* in *E. coli*, for example, can promote β-lactam resistance by upregulating L,D-crosslinking activity of L,D-transpeptidase LdtD (YcbB), presumably due to the increased presence of uncrosslinked glycan strands under these conditions (*Voedts et al., 2021*). It is possible that one function of L,D-TPs is to patch holes in the sacculus via readily available soluble PG strands, constituting an innate 'bike tire repair kit' for the bacterial cell wall. Future studies of this LTG-mediated turnover of uncrosslinked PG could be critical for understanding PG repair mechanisms, as well as general periplasmic homeostasis, both of which have been historically elusive topics.

## Materials and methods

**Key resources table**

| Reagent type (species) or resource | Designation | Source or reference | Identifiers | Additional information |
|---|---|---|---|---|
| Gene (*Vibrio cholerae*) | *mltA* | UniProtKB | *vc_2312* | |
| Gene (*V. cholerae*) | *mltB* | UniProtKB | *vc_1956* | |
| Gene (*V. cholerae*) | *mltC* | UniProtKB | *vc_0450* | |
| Gene (*V. cholerae*) | *mltD* | UniProtKB | *vc_2237* | |

*Continued*

| Reagent type (species) or resource | Designation | Source or reference | Identifiers | Additional information |
|---|---|---|---|---|
| Gene (*V. cholerae*) | *mltF* | UniProtKB | *vc_0866* | |
| Gene (*V. cholerae*) | *mltG* | UniProtKB | *vc_2017* | |
| Gene (*V. cholerae*) | *rlpA* | UniProtKB | *vc_0948* | |
| Gene (*V. cholerae*) | *slt70* | UniProtKB | *vc_0700* | |
| Gene (*V. cholerae*) | *shyA* | UniProtKB | *vc_a0079* | |
| Gene (*V. cholerae*) | *opgH* | UniProtKB | *vc_1287* | |
| Gene (*V. cholerae*) | *ampG* | UniProtKB | *vc_2300* | |
| Gene (*V. cholerae*) | *lacZ* | Kegg2 Gene DB | *vc_2338* | |
| Gene (*Escherichia coli*) | *EcrlpA* | UniProtKB | *b0633* | |
| Gene (*E. coli*) | *EcmltE* | UniProtKB | *b1193* | |
| Gene (*Pseudomonas aeruginosa*) | *ParlpA* | UniProtKB | *pa4000* | |
| Gene (*Bacillus subtilis*) | *sacB* | UniProtKB | bsu34450 | |
| Strain, strain background (*V. cholerae*) | El Tor N16961; wild-type | PMID:243277 | | |
| Recombinant DNA reagent | Δ*mltABCDF* Δ*slt70*; Δ6 LTG | PMID:31286580 | | See *Supplementary file 4*; request from Doerr Lab |
| Recombinant DNA reagent | Δ*mltABCDF* Δ*slt70 mltG::stop*; Δ7 LTG | This study | | See *Supplementary file 4*; request from Doerr lab |
| Recombinant DNA reagent | *mltG::stop* | This study | | See *Supplementary file 4*; request from Doerr Lab |
| Recombinant DNA reagent | pCVD442 | PMID:15109831 | | |
| Recombinant DNA reagent | pTOX5 | PMID:31201277 | Addgene# 127450 | |
| Recombinant DNA reagent | pAM224; pGP704-kanR | PMID:2836362 | | |
| Recombinant DNA reagent | pAM299 | PMID:25631756 | | |
| Recombinant DNA reagent | pJL1 | PMID:24348240 | | |
| Antibody | Anti-mCherry (rabbit polyclonal) | GeneTex | GTX59788 | (1:5000) |
| Antibody | Goat anti-rabbit IRDye 800CW secondary | LI-COR | 926-32211 | (1:16,000) |
| Antibody | Anti-RpoA (mouse monoclonal) | BioLegend | 663104 | (1:10,000) |
| Antibody | Goat anti-mouse IRDye 800CW secondary | LI-COR | 926-32210 | (1:16,000) |
| Chemical compound, drug | Penicillin G potassium salt | Fisher Scientific | CAS: 113-98-4 | |

*Continued on next page*

*Continued*

| Reagent type (species) or resource | Designation | Source or reference | Identifiers | Additional information |
|---|---|---|---|---|
| Chemical compound, drug | Aztreonam | Fisher Scientific | CAS: 78110-38-0 | |
| Chemical compound, drug | Mecillinam | Sigma-Aldrich | CAS: 32887-01-7 | |
| Chemical compound, drug | Cefsulodin sodium salt hydrate | TCI Chemicals | CAS: 1426397-23-0 | |
| Chemical compound, drug | S-(4-Chlorobenzyl) Isothiouronium chloride (MP265) | Chem-Impex International | CAS: 544-47-8 | |
| Chemical compound, drug | Phosphomycin disodium salt (fosfomycin) | Sigma-Aldrich | CAS: 26016-99-9 | |
| Software, algorithm | Fiji | PMID:22743772 | | |
| Software, algorithm | MicrobeJ | PMID:27572972 | | |
| Software, algorithm | Oufti | PMID:26538279 | | |
| Other | BADA | PMID:28989665 | | |

## Bacterial strains and growth conditions

*V. cholerae* strains in this study are derivatives of *V. cholerae* WT El Tor strain N16961 (*Heidelberg et al., 2000*). Construction of plasmids and mutant *V. cholerae* strains is described in the next section along with a table of strains and plasmids used in this work (*Supplementary file 4*).

Strains were grown at 30 or 37°C in Luria-Bertani (LB-Miller, Fisher Bioreagents# BP97235) with or without 1% NaCl, 1% sucrose, or 10% sucrose, or in M9 minimal media + 0.4% glucose (Cold Spring Harbor Protocols) where indicated in the figure legends. Growth media were supplemented with kanamycin (50 μg/mL), ampicillin (25 μg/mL), or chloramphenicol (5 μg/mL) in plates and overnight cultures when needed to maintain plasmids or chromosomal integration of suicide vectors. Genes under $P_{ara}$ and $P_{tac}$ regulation were induced with 0.4% L-arabinose or 200 μM isopropyl-β-D-1-thiolgalactopyranoside (IPTG), respectively.

## Construction of plasmids and strains

A summary of all strains, plasmids, and primers used in this study can be found in *Supplementary file 4*. *E. coli* DH5α *λ pir* was used for general cloning, while *E. coli* SM10 or MFD *λ pir* were used for conjugation into *V. cholerae* (*Ferrières et al., 2010*). Plasmids were constructed using Gibson assembly (*Gibson et al., 2009*) with the exception of plasmids expressing *rlpAΔSPOR* or *rlpA$^{D145A}$*, which were generated by site-directed mutagenesis of the parent wild-type sequence *rlpA* plasmids. All Illumina whole-genome sequencing and variant calling for Δ6 LTG and Δ7 LTG strain verification or suppressor identification were performed by the Microbial Genome Sequencing Center (MiGS, Pittsburg, PA).

Most chromosomal in-frame deletions (or premature stop codon mutants) were generated using the pCVD442 *ampR/sacB* allelic exchange system (*Donnenberg and Kaper, 1991*). 500 bp regions flanking the gene to be deleted were amplified from N16961 genomic DNA by PCR, cloned into suicide vector pCVD442, and conjugated into *V. cholerae*. Conjugation was performed by mixing and pelleting equal volumes of recipient *V. cholerae* and SM10 or MFD *λ pir* donor LB overnight cultures, spotting the mixed pellet onto LB (+600 μM diaminopimelic acid [DAP] for MFD *λ pir*) followed by incubation at 37°C for 3 hr. The first round of selection was performed on LB+ streptomycin (200 μg/mL) + ampicillin (100 μg/mL) at 30°C followed by counterselection on salt-free LB + 10% sucrose + streptomycin at room temperature. Inducers for conditionally essential genes were included in all media during conjugation and selection. Addition of 0.2% glucose was required for maintenance in *E. coli* of plasmids expressing *Vc mltB*. Deletions were verified by PCR.

Introduction of a premature stop codon to *mltG* at its native locus to yield Δ7 LTG as well as in-frame deletions of *opgH* was constructed using the pTOX5 *cmR/msqR* allelic exchange system (*Lazarus et al., 2019*). Flanking regions were cloned into pTOX5 as described for pCVD442.

Conjugation was performed by mixing and pelleting equal parts of recipient *V. cholerae* and donor MFD $\lambda$ *pir*, and spotting onto LB + 1% glucose + 600 µM DAP at 37°C for 5 hr. The first round of selection was performed on LB + chloramphenicol (5 µg/µg mL) + streptomycin + 1% glucose at 30°C. Chloramphenicol-resistant colonies were picked into a 96-well plate containing 200 µL LB + 1% glucose and incubated at 37°C without agitation for 3 hr, then counter selected on LB + 1% rhamnose at 30°C. Mutations were verified by PCR.

Ectopic chromosomal expression from IPTG-inducible $P_{tac}$ was achieved through use of suicide vector pTD101, a pJL1 (*Miyata et al., 2013*) derivative carrying the $P_{tac}$ promoter, a multiple cloning site, and *lacIq* and integrates into the native *V. cholerae lacZ* (*vc2338*) locus. Single genes of interest were amplified from N16961 genomic DNA, introducing a strong consensus RBS (AGGAGA). Genes downstream of *mltG* (*vc2016, vc2015, vc2014*) were amplified together maintaining their native organization, including 30 bp upstream of *vc2016* to retain the native RBS. Selection for double-crossover events was performed as described for pCVD442. Due to hypersensitivity of the Δ7 LTG strain to β-lactams, ampicillin was reduced to 25 µg/mL for the first selection. Due to the osmosensitivity of the Δ7 LTG strain, pTD101 in this strain (and control strains) was maintained as an ampR single crossover without counterselection.

Suicide vector pAM299 (*Möll et al., 2015*) was used to place *rlpA* under $P_{ara}$ control at its native locus for RlpA depletion experiments. pAM299 was introduced via conjugation and selection for single-crossover events on LB + kanamycin (50 µg/mL) + streptomycin. IPTG-inducible overexpression of *rlpA-mCherry* fusions for localization studies and *sacB* for levan toxicity assays was achieved using pHL100 (*Möll et al., 2015*) or its conjugatable derivative pHL100mob. The *sacB* gene was amplified from pCVD442.

## Gene insertion/disruption assay for gene essentiality

The suicide vector pAM224 was used to disrupt genes through single-crossover integration events. 300 bp internal regions in the first third of each respective *orf* (towards the 5′ end) were cloned into pAM224 using methods described above. Quantitative conjugation was performed by washing, mixing, and pelleting 500 µL of recipient *V. cholerae* and donor MFD $\lambda$ *pir*. Pellets were resuspended in 50 µL of LB, spotted onto a 45 µm filter on LB + 600 µM DAP, and incubated at 37°C for 4 hr. Cells were recovered from the filters into 1 mL of LB by vigorous vortexing, and 20 µL of the suspension was reserved for 10-fold serial dilution and spotting onto LB + streptomycin (200 µg/mL) incubated at 30°C to calculate total CFU/mL for all viable *V. cholerae*. The remaining suspension was pelleted and plated on LB + kanamycin (50 µg/mL) + streptomycin incubated at 30°C. Viable CFU/mL were calculated, and kanamycin resistance was verified by patching 50 colonies back onto LB + streptomycin ± kanamycin (all kanR colonies were patched if fewer than 50 colonies were recovered). Transformation efficiency was calculated as a ratio of kanR CFU to all strepR CFU.

## Western blot analysis

Expression of translational mCherry fusions was induced in WT *V. cholerae* with 1 mM IPTG in LB and grown to $OD_{600}$ ~ 0.6. Cells were harvested by centrifugation (9500 × *g*, 15 min) at room temperature and resuspended in 1% SDS + 10 mM dithiothreitol (DTT) lysis buffer. Resuspended cells were incubated at 95°C for 3 min, then sonicated 4 × 5 s at 20% amplitude. Standard Western blots against mCherry were performed using polyclonal mCherry antibody (GeneTex #GTX59788) and detection by IRDye 800CW secondary antibody (LI-COR #926-32211). After imaging for mCherry, the same blots were then reincubated with monoclonal RpoA antibody (BioLegend #663104) detected by IRDye 800CW secondary antibody on an Odyssey CLx imaging device (LI-COR).

## Growth rate experiments

Saturated overnight cultures used for growth curve experiments were washed once and resuspended in final growth media, normalizing to $OD_{600}$ 2.0. Normalized cell suspensions were serially diluted into 200 µL growth media and incubated in a Bioscreen growth plate reader (Growth Curves America) at 37°C with random shaking at maximum amplitude, and $OD_{600}$ recorded at 5 min intervals. Calculations of doublings per hour (DPH) were performed in R as previously described (*Mueller et al., 2019*). Briefly, logarithmic regressions were fitted to sections of growth curves with >5 consecutive values corresponding to $OD_{600}$ 0.03–0.1 (for $10^{-2}$ diluted inocula) or $OD_{600}$ 0.01–0.1 (for $<10^{-3}$ diluted inocula).

Logarithmic regressions of ≥3 replicates with fit value $r^2 > 0.95$ were used to estimate mean DPH for each strain in each growth condition. Growth rates were not calculated for samples that did not reach $OD_{600}$ 0.1 either through absence of detectable growth or lysis prior to reaching $OD_{600}$ 0.1. Growth rates were also not calculated for replicates that were subsequently determined to be suppressors or grew due to irreproducible adaptation. Means within strains between dilution factors and between strains within dilution factors were compared using a two-way ANOVA and Tukey HSD post-hoc test.

## Morphology analysis by microscopy

Strains were grown as described in the figure legends and imaged without fixation on LB 0.8% agarose using a Leica DMi8 inverted microscope. Phase-contrast images were analyzed using MicrobeJ. Default parameter settings were applied, and features (septa) were defined as 25% constriction of cell width. Cell outlines were manually edited as needed.

## Sacculus composition analysis

PG composition from insoluble sacculi samples was analyzed as described previously with some modifications (*Hong, 2016*; *Desmarais et al., 2013*). Briefly, cells were harvested and resuspended in boiled 5% SDS for 1 hr. Sacculi were repeatedly washed by ultracentrifugation (110,000 rpm, 10 min, 20°C) with MilliQ water until SDS was totally removed. Samples were treated with 20 µg Proteinase K (1 hr, 37°C) for removal of Braun's lipoprotein, and finally treated with muramidase (100 µg/mL) for 16 hr at 37°C. Muramidase digestion was stopped by boiling, and coagulated proteins were removed by centrifugation (14,000 rpm, 15 min). For sample reduction, the pH of the supernatants was adjusted to pH 8.5–9.0 with sodium borate buffer and sodium borohydride was added to a final concentration of 10 mg/mL. After incubating for 30 min at room temperature, the sample's pH was adjusted to pH 3.5 with orthophosphoric acid.

UPLC analyses of muropeptides were performed on a Waters UPLC system (Waters Corporation, USA) equipped with an ACQUITY UPLC BEH C18 Column, 130 Å, 1.7 µm, 2.1 mm × 150 mm (Waters, USA) and a dual-wavelength absorbance detector. Elution of muropeptides was detected at 204 nm. Muropeptides were separated at 45°C using a linear gradient from buffer A (formic acid 0.1% in water) to buffer B (formic acid 0.1% in acetonitrile) in an 18 min run, with a 0.25 mL/min flow.

Relative total PG amount was calculated by comparison of the total intensities of the chromatograms (total area) from three biological replicas normalized to the same $OD_{600}$ and extracted with the same volumes. Muropeptide identity was confirmed by MS/MS analysis using a Xevo G2-XS QTof system (Waters Corporation) (see next section for details). Quantification of muropeptides was based on their relative abundances (relative area of the corresponding peak) normalized to their molar ratio. Analyses were performed in biological triplicates, and means were compared with unpaired *t*-tests.

## Soluble peptidoglycan analysis

Sample preparation of soluble PG samples was performed as follows. Bacteria cultures were harvested by centrifugation (4000 rpm, 20 min, 4°C). Cell pellets were gently resuspended and washed twice with ice-cold 0.9% NaCl solution. After pelleting the cells again by centrifugation, they were resuspended in 1 mL water and boiled for 30 min. Samples were centrifuged again to remove cell debris at 14,000 rpm for 15 min, and soluble fractions were transferred to new tubes. Next, samples were filtered using 0.2 µm pore size filters. Half of the sample was treated with muramidase (100 µg/mL) for 16 hr at 37°C. Muramidase digestion was stopped by boiling, and coagulated proteins were removed by centrifugation (14,000 rpm, 15 min). Finally, sample pH was adjusted to pH 3.5 with orthophosphoric acid. When needed, samples were diluted or concentrated by speed vacuum.

Soluble muropeptides were detected and characterized by MS/MS analysis using a Xevo G2-XS QTof system (Waters Corporation) equipped with an ACQUITY UPLC BEH C18 Column (130 Å, 1.7 µm, 2.1 mm × 150 mm; Waters). Muropeptides were separated at 45°C using a linear gradient from buffer A (formic acid 0.1% in water) to buffer B (formic acid 0.1% in acetonitrile) in an 18 min run, with a 0.25 mL/min flow. The QTOF-MS instrument was operated in positive ionization mode. Detection of muropeptides was performed by MS[E] to allow for the acquisition of precursor and product ion data simultaneously using the following parameters: capillary voltage at 3.0 kV, source temperature to 120°C, desolvation temperature to 350°C, sample cone voltage to 40 V, cone gas flow 100 L/hr, desolvation gas flow 500 L/hr, and collision energy (CE): low CE: 6 eV and high CE ramp: 15–40 eV.

Mass spectra were acquired at a speed of 0.25 s/scan. The scan was in a range of 100–2000 *m/z*. Data acquisition and processing was performed using UNIFI software package (Waters Corporation).

An in-house compound library built in UNIFI was used for detection and identification of muropeptides. Subsequent identification and confirmation of each muropeptide were performed by comparison of the retention times and mass spectrometric data to known samples. Quantification was performed by integrating peak areas from extracted ion chromatograms of the corresponding *m/z* value of each muropeptide and normalized to their molar ratio. Soluble muropeptide analyses were performed in biological triplicates, and means were compared with unpaired *t*-tests.

### Antibiotic sensitivity

For zone of inhibition assays, a lawn of saturated overnight culture was spread on an LB plate and allowed to dry for 15 min. Filter disks (6 mm) were dropped with 5 µL of antibiotic solutions (*Supplementary file 4*) onto the lawns and incubated at 37°C for 24 hr before measurements. Means were compared with a one-way ANOVA and Tukey post-hoc test.

### Fluorescent D-amino acid PG labeling

Saturated overnight cultures were diluted 1:100 into LB with 100 µM BADA (*Hsu et al., 2017*) at 37°C for 1.5 hr before addition of antibiotic. Labeled samples were washed one time in LB before imaging on LB 0.8% agarose pads using a Leica DMi8 inverted microscope set for 490 nm excitation.

## Acknowledgements

We would like to acknowledge Dr. Stephen Zinder (Cornell University) for informative comments based on his unique insight into bacterial physiology. We thank Dr. John Helmann (Cornell University) and members of the Dörr lab for helpful comments on the manuscript. Research on autolysins in the Dörr lab is funded by NIH R01 GM130971. Research in the Cava lab is supported by MIMS, the Knut and Alice Wallenberg Foundation (KAW), the Swedish Research Council and the Kempe Foundation. Research in the Van Nieuwenhze lab is supported by the NIH through R01 GM113172 and R35 GM136365.

## Additional information

### Funding

| Funder | Grant reference number | Author |
|---|---|---|
| National Institutes of Health | R01-GM130971 | Tobias Dörr |
| Molecular Infection Medicine Sweden | MIMS2012 | Felipe Cava |
| Knut och Alice Wallenbergs Stiftelse | KAW2012.0184 | Felipe Cava |
| Swedish Research Council | VR2018-02823 | Felipe Cava |
| Kempe Foundation | SMK2062 | Felipe Cava |
| National Institutes of Health | R01-GM113172 | Michael S van Nieuwenhze |
| National Institutes of Health | R35-GM136365 | Michael S van Nieuwenhze |

The funders had no role in study design, data collection and interpretation, or the decision to submit the work for publication.

### Author contributions

Anna Isabell Weaver, Conceptualization, Data curation, Formal analysis, Investigation, Methodology, Validation, Visualization, Writing - original draft, Writing - review and editing; Laura Alvarez, Data

curation, Formal analysis, Investigation, Methodology, Validation, Writing - original draft, Writing - review and editing; Kelly M Rosch, Garrett Sean Wang, Investigation, Validation; Asraa Ahmed, Formal analysis, Investigation, Validation; Michael S van Nieuwenhze, Resources; Felipe Cava, Funding acquisition, Resources, Supervision, Writing - review and editing; Tobias Dörr, Conceptualization, Formal analysis, Funding acquisition, Investigation, Methodology, Project administration, Resources, Supervision, Validation, Writing - original draft, Writing - review and editing

### Author ORCIDs

Anna Isabell Weaver http://orcid.org/0000-0002-0556-0336
Laura Alvarez http://orcid.org/0000-0003-2429-7542
Kelly M Rosch http://orcid.org/0000-0002-6416-1730
Felipe Cava http://orcid.org/0000-0001-5995-718X
Tobias Dörr http://orcid.org/0000-0003-3283-9161

### Decision letter and Author response

Decision letter https://doi.org/10.7554/eLife.73178.sa1
Author response https://doi.org/10.7554/eLife.73178.sa2

## Additional files

### Supplementary files

• Supplementary file 1. Complete sacculus muropeptide composition profiles of lytic transglycosylase (LTG)-deficient mutants. Supports *Figure 1—figure supplement 1D*. Presents mean relative muropeptide abundance ± standard deviation of the mean for all detectable muropeptide species from muramidase-treated sacculi isolated by ultracentrifugation from LTG-deficient mutant cultures during exponential and stationary growth phases. n = 3.

• Supplementary file 2. Complete muropeptide detection profiles of soluble peptidoglycan (PG) material from lytic transglycosylase (LTG)-deficient mutants. Supports *Figure 5B–E*. Presents mean mass peak area ± standard deviation of the mean for all soluble, detectable muropeptide species from muramidase-treated and untreated lysates of LTG-deficient mutant cultures during exponential and stationary growth phases. n = 3.

• Supplementary file 3. Complete muropeptide detection profiles of soluble peptidoglycan (PG) material from ShyA-depleted mutants. Supports *Figure 5F*. Presents mean mass peak area ± standard deviation of the mean for all soluble, detectable muropeptide species from muramidase-treated and untreated lysates of cultures grown with and without ShyA induction. n = 3.

• Supplementary file 4. Bacterial strains, plasmids, and primers used in this study.

• Transparent reporting form

### Data availability

All data generated or analyzed during this study are included in the manuscript and supporting files; Source Data files have been provided for Figures 1 and 3.

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
