## [Editor Report]

This study addresses a major missing element in the understanding of how bacteria grow their cell wall and the role of lytic transglycosylases in this process. It had been previously assumed these enzymes cut glycan strands to make room for the insertion of new glycans. However, results presented in this manuscript demonstrate these enzymes have a very different, yet essential role in degrading uncrosslinked glycan strands in the periplasm. The authors further demonstrate that in the absence of lytic transglycosylases cells undergo periplasmic stress due a toxic accumulation of these ‘free strands’ in the periplasm. The work will be of interest to those in the bacterial growth and division field.

---

## [Decision Letter]

**Decision letter after peer review:**

Thank you for submitting your article "Lytic transglycosylases mitigate periplasmic crowding by degrading soluble cell wall turnover products" for consideration by *eLife*. Your article has been reviewed by 3 peer reviewers, and the evaluation has been overseen by a Reviewing Editor and Gisela Storz as the Senior Editor. The following individual involved in review of your submission has agreed to reveal their identity: Ethan Garner (Reviewer #2).

Essential revisions:

1. Whilst the study is well executed, direct evidence that indeed soluble macromolecular peptidoglycan is being accumulated in the periplasm is missing. Figure 5 presents indirect and qualitative data only. The fluorescent data with BADA labelling could be used to quantify the levels of accumulation in the periplasm. This can be easily done using the existing images by quantifying the fluorescence per cell in each condition/mutant. However, this is still indirect as there is no proof that the BADA accumulation occurs due to incorporation and retention in soluble macromolecular peptidoglycan. If the authors can provide further evidence, this will strengthen their case. Alternatively, if possible, the authors can perform radiolabeling of the peptidoglycan and show that there is more of soluble radiolabeled peptidoglycan per cell in the mutants. If such experiments are not possible, fluorescence microscopy will be acceptable.

2. The authors show a catalytic mutant of RlpA is unable to sustain growth as the only LTG in the cell. However, their wording around RlpA in general is confusing. In the text they note that their δ_7 mutant, which encodes RlpA, 'has no highly active LTGs' (lines 130-131). Does that imply that RlpA is not an LTG? In the discussion they note that *E. coli* RlpA has no LTG activity. Is this enzyme known to have LTG activity in *V. cholerae*? One important control would be to show that the catalytically inactive protein is stable (i.e. that the defect is not due to protein misfolding). This could be supported by looking at protein stability via Western or even quantifying the fluorescence data in Figure S3b

Other points that must be addressed:

1. Figure 5. The authors only present the data for M4N and M4. Does that mean that the soluble peptidoglycan is exclusively composed of polymerized M4? Aren't there any other structures such M5 or M3 or dimers?

2. It would have been also important to show that the accumulation occurs under the other conditions tested such as during β-lactam treatment (Figure 4) but not with moenomycin or Fosfomycin, and that the threshold is reduced when sacB is expressed in the presence of sucrose. Can the authors generate such data?

3. Given that the LTGs are conserved in both gram-negative and positive bacteria, the proposed function of the LTGs becomes a bit confusing, as uncrosslinked strands made by gram-positive bacteria should really diffuse out of the periplasmic space. Thus, it would be beneficial if the authors discuss the implications of these findings (and their model) in light of gram-positives and the conservation of LTGs

4. Figure S2 shows that, even in WT cells, the induction of RPLA seems to give an advantage under spot dilutions, a point not noted in the text. Can the authors explain this, or at least give their hypothesis why this occurs?

5. It would help to define in the figure legend of 3a that this is low salt media (most readers will not notice or understand "LB0N").

6. S3b-d could be better explained in the text – The localization studies with this mutant are stated in a very summarized way; by reading the text, it just appears to be based on previous studies. Thus 1-2 sentences referring to the localization studies would be helpful.

7. In figure 4, MP265 is labeled a GT inhibitor. This is not technically correct and should be corrected. Maybe "rod complex inhibitor?"

8. Figures 4b and 4c should be better explained. Currently, it is stated that "we visualized the fate of PG during antibiotic treatment using the cell wall label BADA." While that statement would make sense, the figure shows spheroplasts, which is not made clear in the figure legend of this figure or the supplementary figures. Adding in a small amount of experimental detail would help the reader understand the experiment (that spheroplast allows discrimination of where the FDAAs are, in the cell wall vs. the periplasm).

9. In general, the manuscript may benefit from some textual changes. Several experiments, or the rationale behind experiments could be explained further. As examples:

In the introduction, the authors do not provide enough background to understand the system. How many LTG are in *V. cholerae*? What is the prevailing model about LTGs? Are these enzymes "misunderstood" or not well understood? Without this context it is hard to understand the nature of the authors' contributions.

Line 165: the authors test if other enzymes can function instead of RlpA in δ_6/7, rather than trying to complement a phenotype of δ_6/7?

Line 120: this assay needs more explanation.

Lines 360-370: the rationale behind this experiment needs more explanation.

10. BADA staining experiments could be more clearly explained. Do the authors stain the cells, treat, and then visualize? Are they then studying the fate of old PG? How does BADA get incorporated into PG in *V. cholerae*? Is it through LDT activity or some other way? Without more explanation, it is hard to interpret the results

11. The authors do a series of experiments showing that delta7 is more susceptible to SacB. What are the data that show sacB produces large polysaccharides molecules in the periplasm rather than (or in addition to) the cytoplasm? This would be important to show as these data are the main test of the model.

12. The authors have other data that all argue for their model that LTG deficient strains have an excess of periplasmic crowding. The suppressor of δ_opgH is intriguing, but does not restore the morphological defects in δ_7, suggesting that the increase in length during prolonged growth may not be caused by periplasmic crowding, or at least is not alleviated by deletion of OpgH. What then does the deletion of OpgH suppress? The low salt experiments do not necessary provide further clarity. The authors indicate that the cells lyse (line 222) but this is not shown anywhere. Growing the cells continually in low salt may not be the hypoosmotic challenge the authors presume. A challenge typically implies an acute change in osmolarity, rather than a prolonged exposure, which may allow cells to adapt. Can the authors provide an explanation?

*Reviewer #1 (Recommendations for the authors):*

This work is well done.

I'm only missing a direct evidence that indeed the amounts of soluble macromolecular peptidoglycan is being accumulated in the periplasm. Figure 5 data is too indirect and is only qualitative. The fluorescent data with BADA labelling could be used to quantify the levels of accumulation. This can be easily done using the existing images by quantifying the fluorescence per cell in each condition/mutant. However, this is still indirect because there is no proof that the BADA accumulation is occurring because of its incorporation and retention in soluble macromolecular peptidoglycan. If the authors can provide these two pieces of evidence, then they will have made their case.

Alternatively, the authors can perform radiolabeling of the peptidoglycan and show that there is more of soluble radiolabeled peptidoglycan per cell in the mutants.

*Reviewer #2 (Recommendations for the authors):*

Overall, I have no significant issues with the paper. Given the importance of this finding to the field and the quality of the work, I definitely believe this paper deserves publication in *eLife*.

*Reviewer #3 (Recommendations for the authors):*

I think this manuscript could be greatly improved with extensive textual changes. Several experiments, or the rationale behind experiments, are not explained in the text and I was confused throughout.

Here are some examples:

In the introduction, the authors do not provide enough background to understand the system. How many LTG are in *V. cholerae*? What is the prevailing model about LTGs? Are these enzymes "misunderstood" or not well understood? Without this context it is hard to understand the nature of the authors' contributions.

Line 165: the authors test if other enzymes can function instead of RlpA in δ_6/7, rather than trying to complement a phenotype of δ_6/7?

Line 120: this assay needs more explanation.

Lines 360-370: the rationale behind this experiment needs more explanation.

---

## [Author Response]

Essential revisions:1. Whilst the study is well executed, direct evidence that indeed soluble macromolecular peptidoglycan is being accumulated in the periplasm is missing. Figure 5 presents indirect and qualitative data only. The fluorescent data with BADA labelling could be used to quantify the levels of accumulation in the periplasm. This can be easily done using the existing images by quantifying the fluorescence per cell in each condition/mutant. However, this is still indirect as there is no proof that the BADA accumulation occurs due to incorporation and retention in soluble macromolecular peptidoglycan. If the authors can provide further evidence, this will strengthen their case. Alternatively, if possible, the authors can perform radiolabeling of the peptidoglycan and show that there is more of soluble radiolabeled peptidoglycan per cell in the mutants. If such experiments are not possible, fluorescence microscopy will be acceptable.

We have now quantified BADA fluorescence in PenG and Fosfomycin-treated cells and added these data to the revised manuscript in supplemental figures S12- S14. While BADA staining is not typically a quantitative assay, the trends support our observations that PenG treatment, not Fosfomycin treatment, is associated with PG accumulation in the LTG-deficient mutants. We would like to point out, however, that our soluble PG analysis in Figure 5 does represent direct evidence of soluble strand accumulation. We directly quantify soluble PG material extracted from cells; the muramidase digest is simply a diagnostic necessity as longer PG strands are not well resolved by standard PG methods and UPLC columns. In this regard, the only possible explanation for the detection of M4 species in soluble PG and whose presence depends on muramidase treatment is that these are not free monomeric muropeptides that belong to uncrosslinked PG strands.

2. The authors show a catalytic mutant of RlpA is unable to sustain growth as the only LTG in the cell. However, their wording around RlpA in general is confusing. In the text they note that their δ_7 mutant, which encodes RlpA, 'has no highly active LTGs' (lines 130-131). Does that imply that RlpA is not an LTG? In the discussion they note that *E. coli* RlpA has no LTG activity. Is this enzyme known to have LTG activity in *V. cholerae*? One important control would be to show that the catalytically inactive protein is stable (i.e. that the defect is not due to protein misfolding). This could be supported by looking at protein stability via Western or even quantifying the fluorescence data in Figure S3b

Alignment of VcRlpA with *P. aeruginosa* RlpA, which has been demonstrated in vivo and in vitro to be an active LTG, suggests VcRlpA retains the active site residues required for PG cleavage. This, as well as the inability of a VcRlpA^D145A^ mutant (based on the alignment with catalytically inactive EcRlpA) to rescue native RlpA depletion from the ∆LTG mutants suggests that VcRlpA is an active LTG and that this activity is required in the absence of all other annotated *V. cholerae* LTGs. We agree that “no highly active LTGs” is confusing and we have changed the text to simply describe the ∆7 LTG mutant as being significantly depleted in LTG activity as measured by anhMurNAc abundance in the sacculus. Lastly, we have conducted Western Blots demonstrating in the revised manuscript that our catalytic site mutant is indeed produced and stable (Figure S3).

Other points that must be addressed:1. Figure 5. The authors only present the data for M4N and M4. Does that mean that the soluble peptidoglycan is exclusively composed of polymerized M4? Aren't there any other structures such M5 or M3 or dimers?

It appears, by our mistake, that all supplemental tables were left out of the original submission. In tables S3-S8, you will find the complete profile of detected soluble muropeptides, including M5 and M3 species. In an effort not to overwhelm the reader, we have demonstrated the soluble strand phenomenon using the most abundant species, M4 and M4N. There is evidence of a muramidase-dependent enrichment of soluble M3 in the ∆6 and ∆7 mutants as well. Reduced dimers were not detected, and anhDimers were in very low abundance, which further suggests that soluble, uncrosslinked single strands accumulate rather than solubilized crosslinked clusters of strands.

2. It would have been also important to show that the accumulation occurs under the other conditions tested such as during β-lactam treatment (Figure 4) but not with moenomycin or Fosfomycin, and that the threshold is reduced when sacB is expressed in the presence of sucrose. Can the authors generate such data?

This is an excellent suggestion and we have indeed started such experiments. However, these analyses are very involved and part of a separate ongoing inquiry into the mechanism of action of β lactam antibiotics and we prefer not to disclose our preliminary data here. We do not believe that these are important experiments to support our conclusions, since in this study we focus on PG homeostasis under physiological conditions (Figure 5), rather than conditions of antibiotic-mediated cell wall stress. We have used cell wall synthesis inhibitors (Figure 4) solely as a tool, i.e. a condition known to exacerbate periplasmic strand accumulation.

3. Given that the LTGs are conserved in both gram-negative and positive bacteria, the proposed function of the LTGs becomes a bit confusing, as uncrosslinked strands made by gram-positive bacteria should really diffuse out of the periplasmic space. Thus, it would be beneficial if the authors discuss the implications of these findings (and their model) in light of gram-positives and the conservation of LTGs

We propose that LTGs and/or closely related hydrolases have at least two separable essential functions: As PG release factors (as described previously), and, newly described here, as mitigators of periplasmic crowding. There are at least three possible scenarios for the involvement of LTGs in Gram-positive bacteria: (1) Only PG release activity is essential in Gram-positives, (2) LTGs are non-essential in Gram-positive bacteria, (3) “periplasmic” crowding actually does happen in Gram-positive bacteria *B. subtilis* PG, for example, is very long-chained and it is conceivable that PG actually constitutes a reasonable diffusion barrier against such long strands. We have discussed these ideas in the revised manuscript.

4. Figure S2 shows that, even in WT cells, the induction of RPLA seems to give an advantage under spot dilutions, a point not noted in the text. Can the authors explain this, or at least give their hypothesis why this occurs?

This might be a misunderstanding: RlpA overexpression does not induce a growth advantage in the WT. Perhaps the reviewer is referring to the fact that the experiment in Figure S2 represents a growth experiment, where the wt simply proliferates between 0 and 3 hours? This has also been clarified in the text.

5. It would help to define in the figure legend of 3a that this is low salt media (most readers will not notice or understand "LB0N").

The media conditions in legend 3a have been clarified.

6. S3b-d could be better explained in the text – The localization studies with this mutant are stated in a very summarized way; by reading the text, it just appears to be based on previous studies. Thus 1-2 sentences referring to the localization studies would be helpful.

The text has been re-written to better separate novel characterization of *V. cholerae* RlpA in this study from previous studies of RlpA from *E. coli* and *P. aeruginosa* as well as emphasize the localization studies of the mCherry fusions.

7. In figure 4, MP265 is labeled a GT inhibitor. This is not technically correct and should be corrected. Maybe "rod complex inhibitor?"

MP265 has been re-labeled as a “Rod complex inhibitor” in Figure 4.

8. Figures 4b and 4c should be better explained. Currently, it is stated that "we visualized the fate of PG during antibiotic treatment using the cell wall label BADA." While that statement would make sense, the figure shows spheroplasts, which is not made clear in the figure legend of this figure or the supplementary figures. Adding in a small amount of experimental detail would help the reader understand the experiment (that spheroplast allows discrimination of where the FDAAs are, in the cell wall vs. the periplasm).

A brief explanation of *V. cholerae’s* tolerance of cell wall acting antibiotics*,* namely the formation of stable, structural PG-deficient spheroplasts, has been added to the description of these experiments. Indeed, the diffuse pattern of signal does additionally suggest that the BADA is labeling soluble PG rather than a peripheral layer of intact PG. Preparation of spheroplasts for microscopy has also been clarified in the legends for Figure 4, S12, and S13.

9. In general, the manuscript may benefit from some textual changes. Several experiments, or the rationale behind experiments could be explained further. As examples:In the introduction, the authors do not provide enough background to understand the system. How many LTG are in *V. cholerae*? What is the prevailing model about LTGs? Are these enzymes "misunderstood" or not well understood? Without this context it is hard to understand the nature of the authors' contributions.

We have made additions to the introduction clarifying the known or proposed roles of LTGs. Our main point is that LTGs are both poorly-understood (no essential physiological function had been assigned prior to this study) and also misunderstood (as in the PG field they are often lumped in with endopeptidases as “space-making” enzymes that promote PG synthesis). In this study, we outline for the first time a reason for why LTG activity is collectively essential.

Line 165: the authors test if other enzymes can function instead of RlpA in δ_6/7, rather than trying to complement a phenotype of δ_6/7?

We would argue that this is essentially what we do with our complementation experiments depicted in Figure 1C. Under conditions without arabinose but with IPTG, RlpA is depleted and growth thus exclusively sustained by an informative panel of other LTGs. We have not attempted to reconstruct a ∆7 with a different LTG as the sole remaining enzyme yet.

Line 120: this assay needs more explanation.

We have added additional explanations.

Lines 360-370: the rationale behind this experiment needs more explanation.

We have added further explanation in the revised manuscript.

10. BADA staining experiments could be more clearly explained. Do the authors stain the cells, treat, and then visualize? Are they then studying the fate of old PG? How does BADA get incorporated into PG in *V. cholerae*? Is it through LDT activity or some other way? Without more explanation, it is hard to interpret the results

BADA does get incorporated through either LDT or PG synthesis activity in *V. cholerae*, but for these experiments, the specific incorporation pathway is inconsequential, since we only focus on the end product (stained PG). We think that what we visualize is not the fate of old PG (otherwise we would see similar strong stains with Fosfomycin, which inhibits cell wall synthesis upstream of PG strand generation by PBPs/SEDS), but rather visualizes the generation of long, uncrosslinked PG strands due to the inhibition of PBP transpeptidase activity. We have added more explanations of this assay to the revised manuscript.

11. The authors do a series of experiments showing that delta7 is more susceptible to SacB. What are the data that show sacB produces large polysaccharides molecules in the periplasm rather than (or in addition to) the cytoplasm? This would be important to show as these data are the main test of the model.

In native *B. subtilis* as well as in *E. coli,* SacB has a canonical Sec signal peptide which is annotated as being cleaved after residue Ala29 (Uniprot G3CAF6_BACIU) to be released extracellularly. A reference (Pereira, et al., 2001) has been added in support of SacB functioning extracellularly and not in the cytoplasm of its native host, *B. subtilis*.

12. The authors have other data that all argue for their model that LTG deficient strains have an excess of periplasmic crowding. The suppressor of δ_opgH is intriguing, but does not restore the morphological defects in δ_7, suggesting that the increase in length during prolonged growth may not be caused by periplasmic crowding, or at least is not alleviated by deletion of OpgH. What then does the deletion of OpgH suppress? The low salt experiments do not necessary provide further clarity. The authors indicate that the cells lyse (line 222) but this is not shown anywhere. Growing the cells continually in low salt may not be the hypoosmotic challenge the authors presume. A challenge typically implies an acute change in osmolarity, rather than a prolonged exposure, which may allow cells to adapt. Can the authors provide an explanation?

We do not fully understand the role of OpgH, but here is our working model: LTGs have at least two essential functions – (1) PG release and (2) mitigating periplasmic crowding, either or both of which can become more important based on osmotic conditions. Since MltG seems to be the main PG release factor (at least based on *E. coli*), which can be partially supplanted by collective action of other LTGs, the ∆7 suffers from both PG release defects and periplasmic crowding defects, perhaps more so in an osmotically challenging low salt medium. The evidence for lysis is that at high inoculum (10^-2^) the ∆7 LTG mutant does grow for a short time, but then we observe a drop in OD_600_, indicative of lysis. According to our model, ∆6, on the other hand, which still has MltG, likely suffers only (or mostly) from a periplasmic crowding defect. Deleting periplasmic glucans only mitigates periplasmic crowding (and probably only partially), which does not help the more defective ∆7, which additionally suffers from lack of the postulated second activity.

The reviewers raise an interesting point regarding the word “challenge”. We indeed specifically make the point that this is not an acute challenge, but rather accumulating damage during prolonged growth, even in salt-free LB. We have thus removed the word “challenge” from the revised manuscript. Importantly, we only use the ∆opgH suppression phenotype as one of many puzzle pieces for our conclusion. The key assay is the direct demonstration of periplasmic soluble PG strands accumulating in both WT and, to a higher degree, the ∆6 LTG mutant (Figure 6).